# CRITICAL ATTENTION SCALING IN LONG-CONTEXT TRANSFORMERS

**Shi Chen**[1], **Zhengjiang Lin**[1], **Yury Polyanskiy**[2], **Philippe Rigollet**[1]
[1]Department of Mathematics, MIT
[2]Department of Electrical Engineering and Computer Science, MIT
`{linzj, schen636, yp}@mit.edu`
`rigollet@math.mit.edu`

## ABSTRACT

As large language models scale to longer contexts, attention layers suffer from a fundamental pathology: attention scores collapse toward uniformity as context length $n$ increases, causing tokens to cluster excessively, a phenomenon known as rank-collapse. While *attention scaling* effectively addresses this deficiency by rescaling attention scores with a polylogarithmic factor $\beta_n$, theoretical justification for this approach remains lacking.

We analyze a simplified yet tractable model that magnifies the effect of attention scaling. In this model, attention exhibits a phase transition governed by the scaling factor $\beta_n$: insufficient scaling collapses all tokens to a single direction, while excessive scaling reduces attention to identity, thereby eliminating meaningful interactions between tokens. Our main result identifies the critical scaling $\beta_n \asymp \log n$ and provides a rigorous justification for attention scaling in YaRN and Qwen, clarifying why logarithmic scaling maintains sparse, content-adaptive attention at large context lengths.

## 1 INTRODUCTION

The attention mechanism is a cornerstone of modern transformer architectures on which Large Language Models (LLMs) rely. Mathematically, an attention layer is a nonlinear operator ATT that maps a collection of *tokens* $\{x_1, \ldots, x_n\}$ from $\mathbb{R}^d$ to $\mathbb{R}^d$. This operator is parametrized by three (possibly sparse) $d$ by $d$ matrices $K, Q$, and $V$ and maps $\{x_1, \ldots, x_n\}$ to $\{x'_1, \ldots, x'_n\}$ using the following formula. Define the normalization operator $N(x) = x/\|x\|$ and for any $i = 1, \ldots, n$ define $q_i = QN(x_i)$, $k_i = KN(x_i)$. Then $x'_i = \mathsf{ATT}(x_1, \ldots, x_n)_i$ is defined as

$$x'_i = V \sum_{j=1}^n N(x_j) A_{ij}, \qquad A_{ij} = \frac{e^{a_{ij}}}{\sum_{k=1}^n e^{a_{ik}}}, \tag{1}$$

where the terms $a_{ij} = q_i^\top k_j$ are called *attention scores*.

A recent line of theoretical work has demonstrated that attention acts as a *contractive* operator that tends to cluster tokens together; see Dong et al. (2021); Geshkovski et al. (2024b; 2025); Karagodin et al. (2024); Geshkovski et al. (2024a); Bruno et al. (2025a); Polyanskiy et al. (2025); Chen et al. (2025a); Cowsik et al. (2024); Giorlandino & Goldt (2025); Rigollet (2025). This clustering effect is also known as "rank-collapse" or "token uniformity" and arises because the distribution of attention scores tends to flatten as the sequence length $n$ grows, causing each token to disperse its attention across too many other tokens rather than focusing selectively.

Various practical solutions have been proposed to curb this clustering behavior. In this work, we focus on simple context-length-aware modifications of the attention mechanism following ideas practically implemented as YaRN (Peng et al., 2023), Qwen (Bai et al., 2023), SSMax (Nakanishi, 2025), and SWAN-GPT (Puvvada et al., 2025). These methods employ a straightforward strategy that rescales attention scores $a_{ij}$ by a single poly-logarithmic factor $\beta_n$; see Table 1. Our goal in this paper is to answer the following fundamental question:

*What is the optimal order of magnitude of the $\beta_n$ scaling?*

To address this question, we propose a highly simplified yet completely tractable model for attention. This model exhibits a phase transition governed by the parameter $\beta_n$ as $n \to \infty$: when $\beta_n$ is below a critical threshold, attention becomes overly contractive and collapses all tokens to a single direction, while when $\beta_n$ is too large, attention acts as an identity operator and fails to process information effectively. More precisely, we establish that the critical parameter $\beta_n$ scales as $\log n$, which corroborates the empirical guidelines underlying YaRN, Qwen, SSMax, and SWAN-GPT.

| Method | $\beta_n$ scaling |
|---------|--------------------|
| YaRN | $(\log n)^2$ |
| Qwen | $\log n$ |
| SSMax | $\log n$ |
| SWAN-GPT | $\log n$ |

Table 1: Attention scaling factors for various methods. The standard attention score $\exp(k_i^\top q_j)$ is replaced with $\exp(C\beta_n k_i^\top q_j)$, $C > 0$.

Our work is intimately connected to the recent contributions of Giorlandino & Goldt (2025) and Cowsik et al. (2024), who investigate the contractive effects of attention mechanisms with random key and query matrices $K$ and $Q$ to establish proper initialization schemes for these parameters. A crucial insight from Cowsik et al. (2024) is that analyzing the evolution of symmetric token configurations provides a more mathematically tractable framework compared to the generic input distributions considered in Geshkovski et al. (2025). This symmetric setting, while simplified, captures essential dynamics of the attention mechanism and enables rigorous theoretical analysis; see also Karagodin et al. (2025).

The choice $\beta_n = \gamma \log n$ appears natural in retrospect. As noted in Nakanishi (2025), with such a scaling the attention weights $A_{ij}$ in Equation (1) become

$$A_{ij} = \frac{n^{\gamma a_{ij}}}{\sum_{k=1}^n n^{\gamma a_{ik}}} .$$

To illustrate the resulting dynamics, consider a simplified regime where all attention scores $a_{ij}$ are of order one: specifically, let $a_{ii} = 1$ and $a_{ij} = \rho > 0$ for $i \neq j$. In this setting, the off-diagonal weights satisfy

$$A_{ij} = \frac{n^{\gamma\rho}}{n^\gamma + (n-1)n^{\gamma\rho}} \sim \begin{cases} 1/n & \text{if } \gamma < \frac{1}{1-\rho} \\ 1/n^{\gamma(1-\rho)} & \text{if } \gamma > \frac{1}{1-\rho} \end{cases}$$

This analysis reveals two distinct regimes. When $\gamma$ is small (subcritical regime), attention weights are asymptotically uniform, resulting in diffuse attention that, as we demonstrate below, leads to severe token contraction. Conversely, when $\gamma$ is large (supercritical regime), off-diagonal weights become negligible with respect to the diagonal ones so that the attention mechanism is effectively suppressed.

The critical regime emerges at the phase boundary $\gamma = \frac{1}{1-\rho}$ where attention can concentrate on a sublinear yet nontrivial number of tokens so as to maintain sufficient connections to facilitate information flow from a small set of important tokens. This sparse attention is related to structured attention mechanisms employed in long-context architectures such as Longformer (Beltagy et al., 2020) and SWIN (Liu et al., 2021) which implement a sliding window over $k \ll n$-nearest neighbors but where proximity is measured in terms of token position rather than embedding. Unlike these structurally constrained approaches that rely on fixed positional neighborhoods, the logarithmic scaling enables the attention pattern to be entirely *content-adaptive*, allowing each token to dynamically select its most relevant context based on semantic similarity rather than positional proximity.

Following similar motivations, Giorlandino & Goldt (2025) establish a compelling analogy between attention dynamics and the random energy model from statistical physics (Derrida, 1981). Using the replica method—an analytical heuristic from statistical physics—they identify a phase transition occurring at $\beta_n \sim \sqrt{\log n}$, which differs from the scalings presented in Table 1. This result represents a significant discrepancy from our findings and highlights fundamental differences in modeling assumptions. More specifically, their approach assumes that the attention scores $a_{ij}$ are correlated Gaussian random variables. This assumption effectively induces a random geometry on the token space, where similarity between tokens is treated as fundamentally random. In this sense, their model bears closer resemblance to recent Kuramoto models on random graphs studied in Abdalla et al. (2022); Jain et al. (2025), where the authors investigate the synchronization of oscillators interacting across the edges of a (sparse) Erdős–Rényi random graph with unit edge weights. However,

in the case of Giorlandino & Goldt (2025), the random graph is both directed and dense, with the edge pointing from token $j$ to token $i$ having weight given by

$$A_{ij} = \frac{e^{\beta_n a_{ij}}}{\sum_{k=1}^n e^{\beta_n a_{ik}}} \tag{2}$$

where $a_{ij}$ are Gaussian random variables. While Giorlandino & Goldt (2025) assumes a specific correlation structure between the Gaussian random variables, the phase transition they uncover is expected to be universal within a large class of random matrices including Wigner ones. Crucially though, in such models, the interaction strength $A_{ij}$ is independent of the positional relationship between tokens $i$ and $j$, making this model qualitatively different from standard attention mechanisms where attention is focused on few (or all) of the preceding tokens. For completeness, we refer readers to Appendix D for a derivation of the critical scaling $\beta_n \asymp \sqrt{\log n}$ in the i.i.d. Gaussian score model.

Bruno et al. (2025b) adopt a different approach to studying the regime where $n \to \infty$ and $\beta_n \to \infty$, in a more general setting than ours. By considering various levels of generality for the matrices $K, Q, V$, this work identifies distinct regimes of token dynamics and relates them to the hardmax ($\beta = \infty$) limit. Importantly, the analysis is conducted in the subcritical regime and differs from the present work in focusing on a broader class of models, for which the critical regime has yet to be precisely characterized. We believe that combining the analytical tools developed in both papers could yield a deeper understanding of this critical regime and represents a promising direction for future research.

The remainder of the paper is organized as follows. Section 2 provides a precise mathematical formulation of the phase transition phenomena for the rescaled attention layer. We begin by analyzing token angles and the contractive behavior of tokens under two settings: an idealized but intuitive simplex model (Section 2.1) and a more realistic model with the simplex constraint relaxed (Section 2.2). In both cases, we identify three distinct regimes of the scaling parameter, each leading to qualitatively different contrastive behaviors of the self-attention layer. Section 2.3 turns to the gradient norm of the rescaled attention operator. Because rank collapse is often accompanied by vanishing gradients, we characterize the gradient dynamics across scaling regimes and show when gradients vanish, or stabilize to non-trivial limits. Section 3 presents our numerical experiments, which validate these theoretical predictions.

Throughout this paper, when we denote a quantity as $o_n(1)$, where $n$ is the number of tokens, we mean there are positive constants $C_1, C_2$ independent of the dimension $d$, such that $|o_n(1)| \leq C_1 n^{-C_2}$. The constants $C_1, C_2$ depend on the assumptions in theorems.

## 2 A PHASE TRANSITION FOR ATTENTION

In this section, we establish the main theorem of this paper, namely a phase transition for the contractive properties of the attention layer when $\beta_n = \gamma \log n$ for some $\gamma > 0$.

Following Geshkovski et al. (2025), we study a simplified version of the attention layer with pre-layer norm that is described in the introduction by assuming that $K = Q = V = I_d$. More specifically, the model we study is given as follows.

For any two points $x, y \in \mathbb{R}^d$, let $\langle x, y \rangle = x^\top y$ denote the standard Euclidean inner product in $\mathbb{R}^d$, and $\|x\| = \sqrt{\langle x, x \rangle}$. Finally, recall that $N(x) := x / \|x\|$.

For any collection of tokens $\{x_1, \ldots, x_n\}$ in $\mathbb{R}^d$, define $y_i = N(x_i) \in \mathbb{S}^{d-1}$ for $i = 1, \ldots, n$ and

$$Z_i := \sum_{k=1}^n e^{a_{ik}}, \qquad A_{ij} := \frac{e^{a_{ij}}}{Z_i}, \qquad a_{ij} := \beta \langle y_i, y_j \rangle, \tag{3}$$

for $i, j = 1, \ldots, n$. We then define

$$\mathsf{ATT}(y_i) := \sum_{j=1}^n A_{ij} y_j. \tag{4}$$

Since the seminal work of He et al. (2016), residual connections are added to modern architectures and naturally act as a regularization scheme of the attention map towards the identity; see Chen et al. (2025b). With said residual connections, each token $x_i$ is mapped to $x_i'$ using the following update rule

$$x_i' := \mathsf{ATT}(y_i) + \alpha x_i, \qquad \alpha \geq 0. \tag{5}$$

Our first goal is to understand where the angle $\angle(x_i', x_j')$ compares to $\angle(x_i, x_j)$. If $\angle(x_i', x_j') < \angle(x_i, x_j)$—or equivalently $\langle y_i', y_j' \rangle > \langle y_i, y_j \rangle$, with $y_i' = N(x_i')$—we say that attention is *contractive*.

The nonlinear update rule (5) can produce complex dynamics, in which some pairs of tokens move closer together while others drift apart. This diversity of motion is in fact the most desirable outcome in practice, and it emerges precisely at the phase transition identified in this study. Beyond this critical regime, the tokens exhibit an unexpectedly cohesive behavior. To delineate the boundaries of the critical regime, we assume that the size and relative positions of the initial tokens are governed by constants independent of the number $n$ of tokens. As an analytically tractable extreme of this assumption, we first consider the case in which the tokens form a regular simplex in $\mathbb{R}^d$ as in Cowsik et al. (2024). Despite its symmetry, this configuration is sufficient to capture and predict the onset of the phase transition. We subsequently relax this constraint in Section 2.2 to show that the same phase transition occurs in more realistic configurations.

## 2.1 THE SIMPLEX CASE

The following assumption was made in Cowsik et al. (2024) and subsequently in Giorlandino & Goldt (2025). While rather stringent—in particular, it requires $d \geq n$—it turns out to provide a tractable yet predictive setup to study the contractive properties of attention.

**Assumption 1** *There exists nonnegative constants $q \geq 0$ and $\rho \in (0, 1)$ such that $\|x_i\|^2 = q$ and $\langle y_i, y_j \rangle = \rho$, for any $i, j = 1, \dots, n$ and $i \neq j$.*

Under Assumption 1, it is easy to see that there are positive constants $\rho'$ and $q'$ such that $\langle y_i', y_j' \rangle = \rho'$ for all $i \neq j$ and $\|x_i'\|^2 = q'$ for all $i$. This simplification gives rise to a tractable phase transition.

**Theorem 2.1** *Under Assumption 1, there is a $\rho' \in (0, 1)$ such that $\langle y_i', y_j' \rangle = \rho'$ for all $i \neq j$. Moreover, if $\beta = \gamma \log n$ where $\gamma$ is a positive constant, then for any $i \neq j$, it holds*

$$\lim_{n \to +\infty} \langle y_i', y_j' \rangle = \begin{cases} \frac{\rho(\alpha\sqrt{q}+1)^2}{\alpha^2 q + 2\alpha\sqrt{q}\rho + \rho} & \text{if } \gamma < \frac{1}{1-\rho}, \\ \frac{\rho(\alpha\sqrt{q}+1)^2}{\alpha^2 q + \alpha\sqrt{q}(1+\rho) + \frac{1+3\rho}{4}} & \text{if } \gamma = \frac{1}{1-\rho}, \\ \rho & \text{if } \gamma > \frac{1}{1-\rho}. \end{cases} \tag{6}$$

Note that when $\gamma \leq \frac{1}{1-\rho}$, the right hand sides of Equation (6) are strictly larger than $\rho$ for any $\alpha \geq 0$. In other words, in the critical and subcritical regimes attention is contractive even in the presence of a residual connection. Of course, when $\alpha \to \infty$, the effects of attention dissipates and the limit tends to $\rho$ for all phases. This is expected as the update from $y_i$ to $y_i'$ tends to the identity map, an effect known to mitigate oversmoothing" in residual neural networks; see Chen et al. (2025b).

Note also that for $\alpha = 0$, that is in absence of residual connections, the limit in Equation (6) reduces to

$$\lim_{n \to +\infty} \langle y_i', y_j' \rangle = \begin{cases} 1 & \text{if } \gamma < \frac{1}{1-\rho}, \\ \frac{4\rho}{1+3\rho} & \text{if } \gamma = \frac{1}{1-\rho}, \\ \rho & \text{if } \gamma > \frac{1}{1-\rho}. \end{cases} \tag{7}$$

In the subcritical case, the tokens contract in one step towards a single cluster when $n \to \infty$ while in the supercritical case, their inner product does not change. In fact, a careful inspection of the proof reveals that in this supercritical regime the attention operator converges to the identity as $n \to \infty$. When $\alpha > 0$, the subcritical case is mitigated by the residual connection which prevents token to

collapse to a single point in one step. Nevertheless, this singular behavior reveals a major limitation in the simplex case: since the tokens are equidistant the phase transition reveals an all-or-nothing phenomenon where attention transitions from $A_{ij} \sim 1/n$ so that $\mathsf{ATT}(y_i) = \bar{y} = \frac{1}{n}\sum_{j=1}^n y_j$ for all $i$ to $A_{ij} = \delta_{ij}$ so that $\mathsf{ATT}(y_i) = y_i$ for all $i$. In the next section, we present a similar result Theorem 2.3, where the simplex assumption is relaxed.

Before we end this section, we present the proof for Equation (7) as a special case of Theorem 2.1. The detailed proof for Theorem 2.1 and the later Theorem 2.3 in Section 2.2 is included in Appendix A.

**Proof 1 (Proof of Equation (7))** *In Equation (5), when $\alpha = 0$, we have that $x_i' = \mathsf{ATT}(y_i)$ for each $i = 1, 2, \ldots, n$. In Equation (3), under Assumption 1, we notice that the quantity $\sum_{k=1}^n e^{a_{ik}}$ in the denominator of $A_{ij}$ is independent of the choice of $i$, and equals to $e^\beta + (n-1)e^{\rho\beta}$. Denote this as $Z := e^\beta + (n-1)e^{\rho\beta}$. Then Equation (4) and (5) become*

$$x_i' = \mathsf{ATT}(y_i) = \frac{1}{Z}\left(e^\beta y_i + \sum_{m \neq i} e^{\rho\beta} y_m\right).$$

*Under Assumption 1, a direct computation shows that for any $i = 1, 2, \ldots, n$,*

$$\langle x_i', x_i'\rangle = \frac{1}{Z^2}\left(e^{2\beta} + 2(n-1)\rho e^{(1+\rho)\beta} + (n-1)(1 + (n-2)\rho)e^{2\rho\beta}\right),$$

*and for any two different $i, j = 1, 2, \ldots, n$,*

$$\langle x_i', x_j'\rangle = \frac{1}{Z^2}\left(\rho e^{2\beta} + 2(1 + (n-2)\rho)e^{\beta(1+\rho)} + \left((n-2) + (n^2 - 3n + 3)\rho\right)e^{2\beta\rho}\right).$$

*See also Lemma A.3 and Lemma A.4 for more detailed computations for $\langle x_i', x_i'\rangle$ and $\langle x_i', x_j'\rangle$.*

*For $Z = e^\beta + (n-1)e^{\rho\beta}$, when we let $\beta = \gamma \log n$, we see that $e^\beta = n^\gamma$ and $ne^{\rho\beta} = n^{1+\rho\gamma}$ in $Z$. The largest term in $Z$ then depends on the relation between $\gamma$ and $1 + \rho\gamma$: when $\gamma < \frac{1}{1-\rho}$, $n^{1+\rho\gamma}$ is the largest term; when $\gamma > \frac{1}{1-\rho}$, $n^\gamma$ is the largest term. We then directly get the following three phases for $Z$ from the above arguments:*

$$Z = \begin{cases} (1 + o_n(1)) \cdot ne^{\rho\beta} & \text{if } \gamma < \frac{1}{1-\rho}, \\ (2 + o_n(1)) \cdot e^\beta & \text{if } \gamma = \frac{1}{1-\rho}, \\ (1 + o_n(1)) \cdot e^\beta & \text{if } \gamma > \frac{1}{1-\rho}, \end{cases} \tag{8}$$

*where the terms $o_n(1)$ go to 0 as $n \to +\infty$. Similarly, we can get the the following three phases for $\langle x_i', x_i'\rangle$:*

$$\lim_{n \to +\infty} \langle x_i', x_i'\rangle = \begin{cases} \rho & \text{if } \gamma < \frac{1}{1-\rho}, \\ \frac{1+3\rho}{4} & \text{if } \gamma = \frac{1}{1-\rho}, \\ 1 & \text{if } \gamma > \frac{1}{1-\rho}. \end{cases} \tag{9}$$

*For $\langle x_i', x_j'\rangle$, we always have that $\lim_{n \to +\infty} \langle x_i', x_j'\rangle = \rho$ for $\gamma$ in these three different regimes. Then Equation (7) follows from these two limits because $\langle y_i', y_j'\rangle = \langle x_i'/\|x_i'\|, x_j'/\|x_j'\|\rangle$.*

## 2.2 THE ALMOST-SIMPLEX CASE

In this section, we relax Assumption 1 to allow pairwise angles and lengths to vary slightly. This relaxation makes it possible for tokens to lie in a dimension $d \ll n$. Although the resulting bounds are not as sharp as those obtained under Assumption 1, they demonstrate that the critical scaling $\beta_n = \Theta(\log n)$ is intrinsic and not merely an artifact of a particular geometric construction.

**Assumption 2** *There exist constants $q_1, q_2 \in (0, \infty), \rho_1, \rho_2 \in (0, 1)$ such that $q_1 \leq \|x_i\|^2 \leq q_2$ and $\rho_1 \leq \langle y_i, y_j\rangle \leq \rho_2$, for any $i, j = 1, \ldots, n$ and $i \neq j$. Moreover, $\rho_1 = \langle y_i, y_j\rangle$ for some $i, j$.*

**Remark 2.2** *Assumption 2 already allows near-ties when $\rho_2$ is close to 1, and it can be further generalized to allow multiple exact ties in the top scores. Specifically, one may assume that there exists a fixed $k \in \mathbb{Z}_+$ such that, for each $i \in \{1, \ldots, n\}$, there are at most $k$ indices $j$ with $\langle y_i, y_j \rangle = 1$. Under this setting, all of our main results continue to hold with the same critical scaling order $\log n$ but with different constants depending on $k$. In fact, this setting is a special case of the more general setting discussed in Appendix C, where Assumption 3 partitions the inner products into three ranges, $[\rho_1, \rho_2]$, $[\rho_3, \rho_4]$, and $\{1\}$, with $0 \le \rho_1 < \rho_2 < \rho_3 < \rho_4 < 1$. The phase transition behavior remains of order $\log n$ in that general setting as well. For clarity and readability of the main exposition, we keep Assumption 2 in the main text.*

It is easy to see using standard probabilistic tools that Assumption 2 holds with high probability when the $y_i$'s are independent random vectors uniformly distributed on a half-sphere for example.

**Theorem 2.3** *Under Assumption 2, we have the following phase transition when $\beta = \gamma \log n$ for some fixed $\gamma > 0$.*

*If $\gamma < \frac{1}{1-\rho_1}$, then there is a constant $\varepsilon > 0$ depending on $\alpha, \rho_2, q_1, q_2$, such that*

$$\varliminf_{n \to +\infty} \min_{i \ne j} \langle y_i', y_j' \rangle \ge \rho_1 + \varepsilon > \rho_1, \tag{10}$$

*which implies that the angle between tokens becomes strictly smaller after an attention layer Equation (5).*

*If $\gamma > \frac{1}{1-\rho_2}$, then for any $i \in [\![1, n]\!]$,*

$$\mathsf{ATT}(y_i) = y_i + o_n(1), \text{ and hence } x_i' = y_i + \alpha x_i + o_n(1), \tag{11}$$

*where the term $o_n(1)$ goes to 0 as $n \to +\infty$ with a speed uniform in $i$. Hence, when $\gamma > \frac{1}{1-\rho_2}$, for any two different $i, j \in [\![1, n]\!]$,*

$$\lim_{n \to +\infty} \langle y_i', y_j' \rangle = \langle y_i, y_j \rangle. \tag{12}$$

*which implies that the angle between tokens does not change after an attention layer Equation (5).*

The proof for Theorem 2.3 is included in Appendix A, but the general intuition is similar to the proof for Equation (7) in Section 2.1. As we have seen in that proof, the first step to build up phase transition regimes for $\langle y_i', y_j' \rangle$ is to study the phase transition regimes for $Z_i$ in Equation (3). Adjusting the logarithmic scaling factor $\gamma$ causes different phase transition regimes for $Z_i$ first. When $\gamma$ is small enough, the weights $e^{a_{ik}}$ consisting of $Z_i$ are asymptotically uniform, and each token almost equally interacts with the other tokens. When $\gamma$ is large enough, each token mostly focuses on itself.

Building on this observation, Theorem 2.1 and Theorem 2.3 together demonstrate that $\gamma$ controls the effective interaction range of each token. In particular, we have seen in Theorem 2.1 the existence of the critical regime when $\gamma = \frac{1}{1-\rho}$. In this case, although the tokens continue to contract, their rate of shrinkage is evidently slower than in the subcritical regime, as shown in Equation (6) and Equation (7).

It is hence natural to ask whether further regimes emerge when $\gamma$ is varied between the supercritical and subcritical threshold. Indeed, in Appendix C, we prove the existence of a nontrivial middle phase when $\gamma$ is between the two extrema $\frac{1}{1-\rho_1}$ and $\frac{1}{1-\rho_2}$, under a refined assumption on the distribution of tokens, which allows for a sharper characterization of the transition. Under this refined assumption, Theorem C.2 show the existence of $\gamma_1, \gamma_2$ such that Equation (5) presents three different phases: $\gamma < \gamma_1, \gamma_1 < \gamma < \gamma_2$, and $\gamma > \gamma_2$. In the extreme regimes, when $\gamma < \gamma_1$, each token interacts with almost all the remaining tokens, while when $\gamma > \gamma_2$, each token only focuses on itself, consistent with Theorem 2.3. In the intermediate regime $\gamma_1 < \gamma < \gamma_2$, however, the weights $e^{a_{ik}}$ concentrate on only a small subset of tokens, so that each $Z_i$ and hence the update in Equation (5) is dominated by a few highly relevant interactions. This shows that the logarithmic scaling enables each token to dynamically select its most relevant context.

We conclude by noting that those $o_n(1)$ terms in our theorems satisfy the bound $|o_n(1)| \le C_1 n^{-C_2}$ for some positive constants $C_1, C_2$ that are independent of $d$ (though varying across theorems). As

a result, the simplex configuration (Assumption 1) and the almost simplex configuration (Assumption 2) remains valid under repeated application of the ATT operator up to $\mathrm{poly}(n)$ iterations. In particular, the accumulated error remains negligible at this scale, so our theorems and arguments extend to transformers with many layers.

## 2.3 PROPAGATION OF GRADIENTS UNDER ATTENTION LAYER

In the previous section, we established how attention scaling influences the propagation of token representations, corresponding to running the Transformer in the forward (inference) direction. During training, however, the Transformer is also executed in the *backward* direction to compute gradients via backpropagation (Rumelhart et al., 1986). In this section, we show that a similar phase transition arises in the backward pass: in the subcritical regime—where token representations rapidly collapse in the forward pass—the gradients also collapse, whereas in the supercritical regime they retain their scale. The stability of gradients is a crucial computational consideration that strongly affects a model's ability to be trained effectively. For this reason, several theoretical analyses of gradient dynamics in Transformers have been conducted, albeit without attention scaling; see, for example, Cowsik et al. (2024); Dong et al. (2021); Noci et al. (2022).

Let the input token configuration be denoted by $X(0)$, and let $X(t)$ represent the positions of all tokens at the output of Transformer layer $t$. To compute gradients, one needs to evaluate the end-to-end input–output Jacobian across $L$ layers of the Transformer. By the chain rule, this Jacobian can be expressed as

$$\frac{\partial X(L)}{\partial X(0)} = \frac{\partial X(L)}{\partial X(L-1)} \frac{\partial X(L-1)}{\partial X(L-2)} \cdots \frac{\partial X(1)}{\partial X(0)}.$$

Thus, the end-to-end Jacobian can be obtained by recursively computing and multiplying the layer-wise Jacobians. This procedure is known as the *adjoint method* in dynamical systems theory (Lions, 1971), and as *backpropagation* in the machine learning community.

Our main result shows that when $\beta_n = \gamma \log n$ with subcritical $\gamma$, the typical singular values of $\frac{\partial X(t+1)}{\partial X(t)}$ are close to zero (apart from the contribution of the residual connection). In contrast, for supercritical values of $\gamma$, the contribution of the attention component to the Jacobian is non-trivial and behaves as a normalization map.

We now proceed with formal definitions. For $x \in \mathbb{R}^d$, let $(x)_u$ denote its $u$-th coordinate for $u = 1, 2, \ldots, d$. The concatenation $X = (x_1, x_2, \ldots, x_n) \in \mathbb{R}^{nd}$ represents the configuration of all tokens. The normalization map is defined by

$$\mathcal{N}(X) = \mathcal{N}(x_1, x_2, \ldots, x_n) \coloneqq \big( N(x_1), N(x_2), \ldots, N(x_n) \big), \tag{13}$$

and the attention map by

$$\mathcal{ATT}(Y) = \mathcal{ATT}(y_1, y_2, \ldots, y_n) \coloneqq \big( \mathsf{ATT}(y_1), \mathsf{ATT}(y_2), \ldots, \mathsf{ATT}(y_n) \big), \tag{14}$$

where $\mathsf{ATT}(y_i)$ is defined in equation 4 and $Y = (y_1, \ldots, y_n)$. Under these definitions, the update equation 5 can be written compactly as

$$X' = \mathcal{ATT}(\mathcal{N}(X)) + \alpha X, \tag{15}$$

where $X' = (x'_1, x'_2, \ldots, x'_n)$.

We define the $nd \times nd$ Jacobian matrix as

$$\nabla_X X' \coloneqq \left( \frac{\partial (x'_j)_v}{\partial (x_i)_u} \right)_{(j,v),(i,u)}, \tag{16}$$

for $i, j = 1, \ldots, n$ and $u, v = 1, \ldots, d$. The matrix norm of $\nabla_X X'$ is given by

$$\|\nabla_X X'\|^2 \coloneqq \mathrm{tr}\big[ (\nabla_X X')^\top \nabla_X X' \big] = \sum_{i,j=1}^{n} \sum_{u,v=1}^{d} \left( \frac{\partial (x'_j)_v}{\partial (x_i)_u} \right)^2. \tag{17}$$

Let $\sigma_1, \sigma_2, \ldots, \sigma_{nd}$ denote the singular values of $\nabla_X X'$. Then the normalized Jacobian norm satisfies

$$\frac{1}{nd}\|\nabla_X X'\|^2 = \frac{1}{nd}\sum_{i=1}^{nd}\sigma_i^2, \tag{18}$$

which represents the mean squared singular value of the Jacobian.

Before stating our results on $\frac{1}{nd}\|\nabla_X X'\|^2$, we note that the Jacobian $\nabla_X X'$ can be decomposed into the residual part $\alpha I_{nd}$ and the attention part $\nabla_X(\mathcal{ATT}(\mathcal{N}(X)))$. As shown in Theorems 2.1 and 2.3, the residual component $\alpha I_{nd}$ does not affect the phase transition behavior. Therefore, to streamline the analysis, we focus exclusively on the attention term $\nabla_X(\mathcal{ATT}(\mathcal{N}(X)))$ by setting $\alpha = 0$ in equation 15. The following theorems characterize $\frac{1}{nd}\|\nabla_X X'\|^2$ under this setting.

**Theorem 2.4** *Adopt Assumption 1 and Equation (15) with $\alpha = 0$. Then, we have the following phase transition phenomenon: let $\beta = \gamma \log n$ where $\gamma$ is a positive constant.*

*If $\gamma < \frac{1}{1-\rho}$,*

$$\frac{1}{nd}\|\nabla_X X'\|^2 = 0 + o_n(1). \tag{19}$$

*If $\gamma = \frac{1}{1-\rho}$*

$$\frac{1}{nd}\|\nabla_X X'\|^2 = \frac{1}{4q}\left(1 - \frac{1}{d}\right) + o_n(1). \tag{20}$$

*If $\gamma > \frac{1}{1-\rho}$*

$$\frac{1}{nd}\|\nabla_X X'\|^2 = \frac{1}{q}\left(1 - \frac{1}{d}\right) + o_n(1). \tag{21}$$

*In both cases, the terms $o_n(1)$ go to $0$ as $n \to +\infty$, with speeds depending on $\gamma, \rho, q$.*

The results of the previous theorem show that under the simplex assumption, the phase transition in the backward dynamics (for gradients) is as sharp as for the forward pass: for small $\gamma$, gradients do not flow through the attention block.

We can also extend the analysis for Theorem 2.4 to the relaxed Assumption 2.

**Theorem 2.5** *Adopt Assumption 2 and Equation (15) with $\alpha = 0$. Then, we have the following phase transition phenomenon: let $\beta = \gamma \log n$ where $\gamma$ is a positive constant.*

*If $\gamma < \frac{1}{1-\rho_1}$,*

$$\frac{1}{nd}\|\nabla_X X'\|^2 \leq 4\frac{\gamma^2(\log(n))^2}{q_1 d} + o_n(1), \tag{22}$$

*If $\gamma > \frac{1}{1-\rho_2}$,*

$$\frac{1}{nd}\|\nabla_X X'\|^2 \geq \frac{1}{q_2}\left(1 - \frac{1}{d}\right) + o_n(1), \tag{23}$$

*which is away from $0$ even when $d, n$ is very large. Indeed, when $\gamma > \frac{1}{1-\rho_2}$, for any fixed $i, j \in [\![1, n]\!]$,*

$$\left(\frac{\partial(\mathsf{ATT}(N(x_j)))_v}{\partial(x_i)_u}\right)_{d\times d} = \frac{\delta_{ij}}{\|x_i\|}\left(I_d - y_i y_i^T\right) + \mathbf{o}_n(1) + o_n(1)\cdot I_d, \tag{24}$$

*where the leading order term is exactly $\frac{\partial(N(x_j))_v}{\partial(x_i)_u}$ as shown in Proposition B.1. Here, $I_d$ is the $d \times d$ identity matrix, the term $\mathbf{o}_n(1)$ ($o_n(1)$, respectively) is a $d \times d$ matrix (constant, respectively) with matrix norm as defined in Equation (17) (value, respectively) going to $0$ as $n \to +\infty$, with a speed independent of $i, j$ but only depending on $\gamma, \rho_2, q_1$.*

We present the proofs for Theorem 2.4 and Theorem 2.5 in Appendix B. Note that the $\frac{\log^2 n}{d}$ term in equation 22 is small for typical values of $n$ and $d$ used in Transformers. Theorem 2.4 and Theorem 2.5 also corroborate the fact that tokens collapse fast when $\gamma$ is in the subcritical regime, while each token only focuses on itself when $\gamma$ is in the supercritical regime.

## 3 NUMERICAL EXPERIMENTS

This section reports numerical experiments designed to support our theoretical predictions. In the following numerical experiments, we test the phase transition in the almost-simplex case as Section 2.2. We generate samples $\{x_1, \ldots, x_n\} \subset \mathbb{R}^d$ such that the expectations $\mathbb{E}\|x_i\|^2 = 1$ and $\mathbb{E}\langle x_i, x_j \rangle = \rho \in [0, 1]$ for $i \neq j$. More precisely, we generate $x_i$ according to

$$x_i = \sqrt{\rho}\, z_0 + \sqrt{1-\rho}\, z_i \,, \tag{25}$$

where $z_0, z_1, \ldots, z_n$ are i.i.d. standard Gaussian vectors in $\mathbb{R}^d$. The generated samples satisfy the Assumption 2 with high probability.

In Figure 1, we plot the input-to-output angle ratio $\lambda$, defined as

$$\lambda = \frac{2}{n(n-1)} \sum_{1 \leq i < j \leq n} \frac{1 - \langle y_i', y_j' \rangle}{1 - \langle y_i, y_j \rangle} \,, \tag{26}$$

for samples processed through a single self-attention layer with different $\gamma$ and of different dimensions $d$. Consistent with our theoretical predictions, the layer acts as a contraction mapping when $\gamma$ is small, reducing pairwise output angles, whereas for large $\gamma$ the output angles remain nearly unchanged from the input. Moreover, in the large $d$ regime the angle between input tokens $\langle y_i, y_j \rangle$ ($i \neq j$) concentrate near $\rho$, so that the simplex Assumption 1 is effectively satisfied. In this setting, we observe a sharp phase transition in agreement with Theorem 2.1. In the small $d$ regime, however, the input tokens $\langle y_i, y_j \rangle$ randomly distributed in an interval $(\rho_1, \rho_2)$, and an intermediate phase emerges in which the contraction is only partial: some angles shrink significantly while others remain close to their original values, which smooths out the transition.

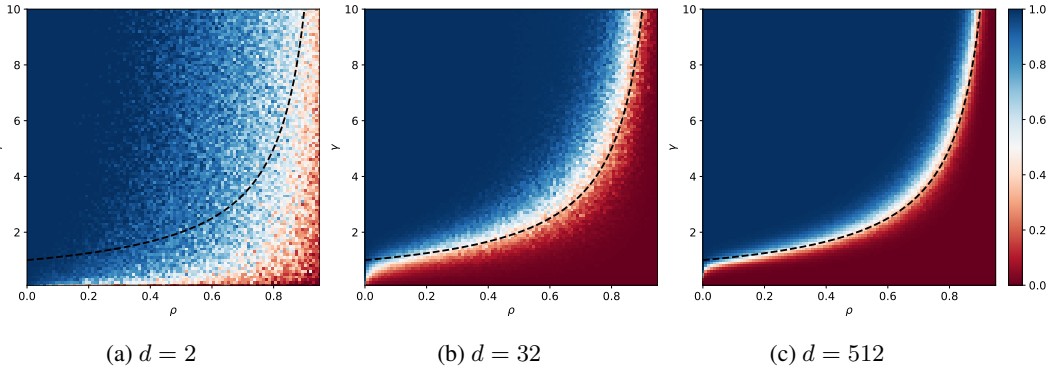

(a) $d = 2$        (b) $d = 32$        (c) $d = 512$

Figure 1: Plots of the input-to-output angle ratio $\lambda$, defined in Equation (26), as a function of $\rho$ and $\gamma$. The tokens are first normalized by a pre-layer normalization and then passed through a single self-attention layer (4), with residual connections and MLP layers omitted. The dashed curve corresponds to $\gamma = \frac{1}{1-\rho}$, which approximates the actual phase transition with increasing accuracy as $d$ grows, as implied by Theorem 2.1.

In Figure 2, we plot the normalized matrix norm for the $nd \times nd$ matrix $\nabla_X X'$, defined as

$$\eta = \frac{1}{nd} \|\nabla_X X'\|^2 \,, \tag{27}$$

for samples passed through a single self-attention layer with varying $\gamma$ and dimension $d$. Across all three plots, the normalized gradient norm remains close to 0 when $\gamma$ is small, while for large $\gamma$ it approaches $1 - 1/d$, consistent with Theorem 2.5. Similar to the token-angle behavior, a sharp phase

transition emerges near $\gamma = \frac{1}{1-\rho}$ in the large-$d$ regime, in agreement with the predictions under the simplex assumption. In lower dimensions, fluctuations in the pairwise angle prevent perfect concentration, and the transition is smoothed into an intermediate regime where the gradient norm only partially stabilizes.

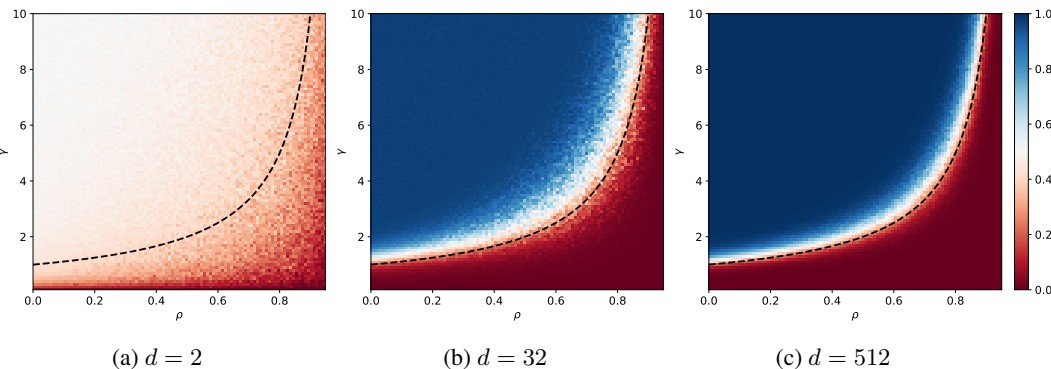

(a) $d = 2$           (b) $d = 32$           (c) $d = 512$

Figure 2: Plots of the normalized norm $\eta$ of the gradient, defined by Equation (27), as a function of $\rho$ and $\gamma$. The tokens are first normalized by a pre-layer normalization and then passed through a single self-attention layer (4), with residual connections and MLP layers omitted. The dash curve shows $\frac{1}{1-\rho}$, which approximate the actual phase transition with increasing accuracy as $d$ grows, as implied by Theorem 2.4. The matrix norm $\eta$ is computed by the Hutchinson trace estimator (Hutchinson, 1989), based on the definition in Equation (17).

## 4    CONCLUSION

This paper develops a framework for understanding phase transitions in self-attention as the context length $n$ grows, identifying a critical scaling $\beta_n \asymp \log n$ that separates a subcritical contractive regime from a supercritical unchanged regime. A central message is that this transition is rooted in the geometry of the score landscape: in our model, the gaps between the top ordered scores remain $O(1)$, which leads to the $\log n$ scaling. We show that this scaling is robust under various perturbations and in settings permitting multiple near-ties, demonstrating that the $\log n$ law is a structural consequence of content-adaptive interactions.

### ACKNOWLEDGMENTS

Philippe Rigollet is supported by NSF grants DMS-2022448.

### LLM USAGE

Large Language Models (LLMs) were used during peer review for grammar and syntax refinement only. All ideas, technical content, analyses and conclusions remain the authors' work.

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

## A    PROOF OF THEOREM 2.1 AND THEOREM 2.3

In this section, we adopt Assumption 2 and prove Theorem 2.3 first. Then, we prove Theorem 2.1. To simplify notations, we define $\llbracket 1, n \rrbracket := \{1, 2, \ldots, n\}$ for any $n \in \mathbb{Z}_+$.

We study the asymptotics of the quantity $\langle x_i', x_j' \rangle$ as $n \to +\infty$. We use the notation

$$Z_i := \sum_{k=1}^{n} e^{a_{ik}} = e^{\beta} + \sum_{k \neq i} e^{a_{ik}}. \tag{28}$$

**Lemma A.1** *Let $\beta = \gamma \log n$ where $\gamma$ is a positive constant. Under Assumption 2 and Equation (5), for any $i \in \llbracket 1, n \rrbracket$,*

$$Z_i = \begin{cases} (1 + o_n(1)) \cdot \left( \sum_{k \neq i} e^{a_{ik}} \right) & \text{if } \gamma < \frac{1}{1-\rho_1}, \\ (1 + o_n(1)) \cdot e^{\beta} & \text{if } \gamma > \frac{1}{1-\rho_2}, \end{cases} \tag{29}$$

*where the terms $o_n(1)$ go to 0 as $n \to +\infty$ with speeds independent of $i$ but only depending on $\gamma, \rho_1, \rho_2$.*

**Proof 2 (Proof of Lemma A.1)** *We notice that*

$$Z_i = e^{\beta} + \sum_{k \neq i} e^{a_{ik}}. \tag{30}$$

*We also notice that $e^{\beta t} = n^{\gamma t}$ for any $t$. It then holds that $e^{\beta} = n^{\gamma}$ and*

$$n^{\gamma \rho_1}(n-1) \leq \sum_{k \neq i} e^{a_{ik}} \leq n^{\gamma \rho_2}(n-1). \tag{31}$$

*Hence, when $\gamma < \frac{1}{1-\rho_1}$, $n^{\gamma} < n^{1+\gamma \rho_1}$, the leading order term in $Z_i$ is $\sum_{k \neq i} e^{a_{ik}}$. We also see that*

$$Z_i = \left( \frac{e^{\beta}}{\left( \sum_{k \neq i} e^{a_{ik}} \right)} + 1 \right) \cdot \left( \sum_{k \neq i} e^{a_{ik}} \right), \tag{32}$$

*with*

$$\frac{e^{\beta}}{\left( \sum_{k \neq i} e^{a_{ik}} \right)} \leq \frac{n^{\gamma}}{n^{\gamma \rho_1}(n-1)}, \tag{33}$$

*which goes to 0 as $n \to +\infty$, and is independent of $i$ but only depending on $\gamma, \rho_1$. Similarly, when $\gamma > \frac{1}{1-\rho_2}$, $n^{\gamma} > n^{1+\gamma \rho_2}$, the leading order term in $Z_i$ is $e^{\beta}$, and similar arguments hold true.*

**Lemma A.2** *Let $\beta = \gamma \log n$ where $\gamma$ is a positive constant. Under Assumption 2 and Equation (5), if $\gamma > \frac{1}{1-\rho_2}$, then for any $i \in \llbracket 1, n \rrbracket$,*

$$\text{ATT}(y_i) = y_i + o_n(1), \text{ and hence } x_i' = y_i + \alpha x_i + o_n(1), \tag{34}$$

*where the term $o_n(1)$ goes to 0 as $n \to +\infty$ with a speed independent of $i$ but only depending on $\gamma, \rho_2$.*

**Proof 3 (Proof of Lemma A.2)** *According to Lemma A.1, we see that when $\gamma > \frac{1}{1-\rho_2}$, $n^\gamma > n^{1+\gamma\rho_2}$, and hence,*

$$\mathsf{ATT}(y_i) = Z_i^{-1}\left(e^\beta y_i + \sum_{j\neq i} e^{a_{ij}} y_j\right) = (1 + o_n(1))\left(y_i + e^{-\beta}\sum_{j\neq i} e^{a_{ij}} y_j\right). \qquad (35)$$

*Because $\|y_j\| = 1$,*

$$\left\|e^{-\beta}\sum_{j\neq i} e^{a_{ij}} y_j\right\| \leq e^{-\beta}\sum_{j\neq i} e^{a_{ij}} \leq n^{-\gamma}\cdot n^{\gamma\rho_2}(n-1), \qquad (36)$$

*which goes to 0 as $n \to +\infty$, and is independent of $i$ but only depending on $\gamma, \rho_2$. This shows that when $\gamma > \frac{1}{1-\rho_2}$,*

$$\mathsf{ATT}(y_i) = (1 + o_n(1))(y_i + o_n(1)) = y_i + o_n(1). \qquad (37)$$

**Lemma A.3** *Under Assumption 2 and Equation (5), for any $i \in [\![1,n]\!]$,*

$$\langle x_i', x_i'\rangle = \alpha^2\|x_i\|^2 + \frac{2\alpha\|x_i\|}{Z_i}\left(e^\beta + \sum_{j\neq i} e^{a_{ij}}\langle y_i, y_j\rangle\right)$$
$$+ \frac{1}{Z_i^2}\left(e^{2\beta} + 2e^\beta\sum_{j\neq i} e^{a_{ij}}\langle y_i, y_j\rangle + \sum_{j\neq i}\sum_{k\neq i} e^{a_{ij}+a_{ik}}\langle y_k, y_j\rangle\right). \qquad (38)$$

*Let $\beta = \gamma \log n$ where $\gamma$ is a positive constant. When $\gamma < \frac{1}{1-\rho_1}$,*

$$\langle x_i', x_i'\rangle = \alpha^2\|x_i\|^2 + 2\alpha\|x_i\|\frac{\sum_{k\neq i} e^{a_{ik}}\langle y_i, y_k\rangle}{\sum_{k\neq i} e^{a_{ik}}} + \frac{\sum_{k\neq i}\sum_{l\neq i} e^{a_{ik}+a_{il}}\langle y_k, y_l\rangle}{\left(\sum_{k\neq i} e^{a_{ik}}\right)^2} + o_n(1). \qquad (39)$$

*When $\gamma > \frac{1}{1-\rho_2}$,*

$$\langle x_i', x_i'\rangle = (\alpha\|x_i\| + 1)^2 + o_n(1). \qquad (40)$$

*In both cases, the terms $o_n(1)$ go to 0 as $n \to +\infty$ with speeds independent of $i$ but only depending on $\gamma, \rho_1, \rho_2, \alpha$.*

**Proof 4 (Proof of Lemma A.3)** *According to Equation (5), we see that*
$$\langle x_i', x_i'\rangle = \alpha^2\|x_i\|^2 + 2\alpha\langle x_i, \mathsf{ATT}(y_i)\rangle + \langle\mathsf{ATT}(y_i), \mathsf{ATT}(y_i)\rangle. \qquad (41)$$
*Equation (38) follows from direct computations. Two phase transitions Equation (39) and Equation (40) follow from similar arguments as in Lemma A.1.*

**Lemma A.4** *Under Assumption 2 and Equation (5), for any two different $i, j \in [\![1,n]\!]$,*

$$\langle x_i', x_j'\rangle = \alpha^2\langle x_i, x_j\rangle + \frac{\alpha\|x_j\|}{Z_i}\left(e^\beta\langle y_j, y_i\rangle + \sum_{k\neq i} e^{a_{ik}}\langle y_j, y_k\rangle\right) + \frac{\alpha\|x_i\|}{Z_j}\left(e^\beta\langle y_i, y_j\rangle + \sum_{l\neq j} e^{a_{jl}}\langle y_i, y_l\rangle\right)$$
$$+ \frac{1}{Z_i Z_j}\left(e^{2\beta}\langle y_i, y_j\rangle + e^\beta\sum_{k\neq i} e^{a_{ik}}\langle y_j, y_k\rangle + e^\beta\sum_{l\neq j} e^{a_{jl}}\langle y_i, y_l\rangle + \sum_{k\neq i}\sum_{l\neq j} e^{a_{ik}+a_{jl}}\langle y_k, y_l\rangle\right). \qquad (42)$$

*Let $\beta = \gamma \log n$ where $\gamma$ is a positive constant. When $\gamma < \frac{1}{1-\rho_1}$,*

$$\langle x_i', x_j'\rangle = \alpha^2\langle x_i, x_j\rangle + \alpha\|x_j\|\frac{\sum_{k\neq i} e^{a_{ik}}\langle y_j, y_k\rangle}{\sum_{k\neq i} e^{a_{ik}}} + \alpha\|x_i\|\frac{\sum_{l\neq j} e^{a_{jl}}\langle y_i, y_l\rangle}{\sum_{l\neq j} e^{a_{jl}}}$$
$$+ \frac{\sum_{k\neq i}\sum_{l\neq j} e^{a_{ik}+a_{jl}}\langle y_k, y_l\rangle}{\left(\sum_{k\neq i} e^{a_{ik}}\right)\cdot\left(\sum_{l\neq j} e^{a_{jl}}\right)} + o_n(1). \qquad (43)$$

*When $\gamma > \frac{1}{1-\rho_2}$,*

$$\langle x_i', x_j' \rangle = (\alpha \|x_i\| + 1)(\alpha \|x_j\| + 1)\langle y_i, y_j \rangle + o_n(1). \tag{44}$$

**Proof 5 (Proof of Lemma A.4)** *According to Equation (5), we see that for two different $i, j \in [\![1, n]\!]$,*

$$\langle x_i', x_j' \rangle = \alpha^2 p + \alpha \langle x_i, \mathsf{ATT}(y_j) \rangle + \alpha \langle x_j, \mathsf{ATT}(y_i) \rangle + \langle \mathsf{ATT}(y_i), \mathsf{ATT}(y_j) \rangle. \tag{45}$$

*Equation (42) follows from direct computations. Two phase transitions Equation (43) and Equation (44) follow from similar arguments as in Lemma A.1.*

Next, we prove Theorem 2.3.

**Proof 6 (Proof of Theorem 2.3)** *We first discuss the case when $\gamma < \frac{1}{1-\rho_1}$. According to Equation (43) and Assumption 2, we see that*

$$\begin{aligned}
\langle x_i', x_j' \rangle &\geq \alpha^2 \|x_i\| \|x_j\| \rho_1 + \alpha \|x_j\| \rho_1 + \alpha \|x_i\| \rho_1 + \rho_1 + o_n(1) \\
&= \rho_1(\alpha \|x_i\| + 1)(\alpha \|x_j\| + 1) + o_n(1).
\end{aligned} \tag{46}$$

*By Equation (39), we see that*

$$\begin{aligned}
\langle x_i', x_i' \rangle &\leq \alpha^2 \|x_i\|^2 + 2\alpha \|x_i\| \rho_2 + \rho_2 + o_n(1) \\
&= \alpha^2 \|x_i\|^2 + 2\alpha \|x_i\| + 1 - (1 - \rho_2)(1 + 2\alpha \|x_i\|) + o_n(1) \\
&\leq (\alpha \|x_i\| + 1)^2 - (1 - \rho_2)(1 + 2\alpha q_1) + o_n(1).
\end{aligned} \tag{47}$$

*We have a similar inequality for $\langle x_j', x_j' \rangle$. So, there is a constant $\delta > 0$ depending on $\rho_2, \alpha, q_1, q_2$ and independent of $n$, such that*

$$\frac{1}{\|x_i'\|} \geq \frac{1 + \delta}{\alpha \|x_i\| + 1} + o_n(1), \text{ and } \frac{1}{\|x_j'\|} \geq \frac{1 + \delta}{\alpha \|x_j\| + 1} + o_n(1). \tag{48}$$

*Hence,*

$$\langle y_i', y_j' \rangle \geq \rho_1(1 + \delta)^2 + o_n(1) \geq \rho_1 + \varepsilon + o_n(1), \tag{49}$$

*for $\varepsilon = \rho_1(1 + 2\delta)\delta > 0$ independent of $n$.*

*For the case when $\gamma < \frac{1}{1-\rho_2}$, Equation (11) and Equation (12) follow directly from Lemma A.2, Lemma A.3, and Lemma A.4.*

**Proof 7 (Proof of Theorem 2.1)** *We notice that Assumption 1 corresponds to the special case when $q_1 = q_2 = q$ and $\rho_1 = \rho_2 = \rho$ in Assumption 2. Clearly, $Z_i$ is independent of the choice of $i \in [\![1, n]\!]$ by its definition Equation (28). According to the explicit forms Equation (38) in Lemma A.3 and Equation (42) in Lemma A.4, one directly sees that both $\langle x_i, x_i \rangle$ and $\langle x_i, x_j \rangle$ are independent of the choices of $i, j \in [\![1, n]\!]$. We can further compute that for any $i \in [\![1, n]\!]$,*

$$\lim_{n \to +\infty} \langle x_i', x_i' \rangle = \begin{cases} \alpha^2 q + 2\alpha\sqrt{q}\rho + \rho & \text{if } \gamma < \frac{1}{1-\rho}, \\ \alpha^2 q + \alpha\sqrt{q}(1 + \rho) + \frac{1+3\rho}{4} & \text{if } \gamma = \frac{1}{1-\rho}, \\ (\alpha\sqrt{q} + 1)^2 & \text{if } \gamma > \frac{1}{1-\rho}, \end{cases} \tag{50}$$

*and for any two different $i, j \in [\![1, n]\!]$,*

$$\lim_{n \to +\infty} \langle x_i', x_j' \rangle = \rho(\alpha\sqrt{q} + 1)^2. \tag{51}$$

*Equation (6) follows from Equation (50) and Equation (51).*

*When $\gamma < \frac{1}{1-\rho}$, we see that*

$$\lim_{n \to +\infty} \langle y_i', y_j' \rangle = \frac{\rho(\alpha\sqrt{q} + 1)^2}{\alpha^2 q + 2\alpha\sqrt{q}\rho + \rho} > \frac{\rho(\alpha\sqrt{q} + 1)^2}{\alpha^2 q + 2\alpha\sqrt{q} + 1} = \rho, \tag{52}$$

*where the strict inequality is because $\rho < 1$. When $\gamma = \frac{1}{1-\rho}$, we can similarly show that $\lim_{n \to +\infty} \langle y_i', y_j' \rangle > \rho$. This completes the proof for Theorem 2.1.*

## B  PROOF OF THEOREM 2.4 AND THEOREM 2.5

We prove Theorem 2.5 first. We need to explicitly compute terms in $\frac{\partial(\mathrm{ATT}(N(x_j)))_v}{\partial(x_i)_u}$, for which we need the following lemmas.

### B.1  PROOF OF THEOREM 2.5

**Lemma B.1** *For any $i, k \in [\![1, n]\!]$ and $u, w \in [\![1, d]\!]$,*

$$\frac{\partial(N(x_k))_w}{\partial(x_i)_u} = \delta_{ik} \frac{\delta_{wu}\|x_k\|^2 - (x_k)_w(x_k)_u}{\|x_k\|^3}. \tag{53}$$

**Proof 8 (Proof of Lemma B.1)**

$$\frac{\partial(N(x_k))_w}{\partial(x_i)_u} = \frac{\partial((x_k)_w \cdot \|x_k\|^{-1})}{\partial(x_i)_u} = \delta_{ik} \frac{\delta_{wu}\|x_k\| - (x_k)_w \cdot \frac{(x_k)_u}{\|x_k\|}}{\|x_k\|^2}. \tag{54}$$

**Lemma B.2** *For any $k, j \in [\![1, n]\!]$ and $w, v \in [\![1, d]\!]$,*

$$
\begin{aligned}
&\frac{\partial(\mathrm{ATT}(y_j))_v}{\partial(y_k)_w} \\
&= \Bigg[ \left( \delta_{kj}\beta \left( \sum_{m=1}^{n} e^{\beta\langle y_j, y_m\rangle}(y_m)_w(y_m)_v \right) + e^{\beta\langle y_j, y_k\rangle}(\beta(y_j)_w(y_k)_v + \delta_{wv}) \right) \cdot \left( \sum_{l=1}^{n} e^{\beta\langle y_j, y_l\rangle} \right) \\
&\quad - \left( \delta_{kj}\beta \left( \sum_{l=1}^{n} e^{\beta\langle y_j, y_l\rangle}(y_l)_w \right) + \beta e^{\beta\langle y_j, y_k\rangle}(y_j)_w \right) \cdot \left( \sum_{m=1}^{n} e^{\beta\langle y_j, y_m\rangle}(y_m)_v \right) \Bigg] \\
&\quad \cdot \left( \sum_{l=1}^{n} e^{\beta\langle y_j, y_l\rangle} \right)^{-2}.
\end{aligned}
\tag{55}
$$

**Proof 9 (Proof of Lemma B.2)** *By Equation (4),*

$$(\mathrm{ATT}(y_j))_v = \frac{\sum_{m=1}^{n} e^{\beta\langle y_j, y_m\rangle}(y_m)_v}{\sum_{l=1}^{n} e^{\beta\langle y_j, y_l\rangle}}. \tag{56}$$

*A direct computation shows that*

$$
\begin{aligned}
&\frac{\partial(\mathrm{ATT}(y_j))_v}{\partial(y_k)_w} \\
&= \Bigg[ \left( \sum_{m=1}^{n} (\delta_{kj}\beta(y_m)_w(y_m)_v + \delta_{km}\beta(y_j)_w(y_m)_v + \delta_{km}\delta_{wv}) e^{\beta\langle y_j, y_m\rangle} \right) \cdot \left( \sum_{l=1}^{n} e^{\beta\langle y_j, y_l\rangle} \right) \\
&\quad - \left( \sum_{l=1}^{n} (\delta_{kj}(y_l)_w + \delta_{kl}(y_j)_w) \beta e^{\beta\langle y_j, y_l\rangle} \right) \cdot \left( \sum_{m=1}^{n} e^{\beta\langle y_j, y_m\rangle}(y_m)_v \right) \Bigg] \\
&\quad \cdot \left( \sum_{l=1}^{n} e^{\beta\langle y_j, y_l\rangle} \right)^{-2} \\
&= \Bigg[ \left( \delta_{kj}\beta \sum_{m=1}^{n} e^{\beta\langle y_j, y_m\rangle}(y_m)_w(y_m)_v + e^{\beta\langle y_j, y_k\rangle}(\beta(y_j)_w(y_k)_v + \delta_{wv}) \right) \cdot \left( \sum_{l=1}^{n} e^{\beta\langle y_j, y_l\rangle} \right) \\
&\quad - \left( \delta_{kj}\beta \sum_{l=1}^{n} e^{\beta\langle y_j, y_l\rangle}(y_l)_w + \beta e^{\beta\langle y_j, y_k\rangle}(y_j)_w \right) \cdot \left( \sum_{m=1}^{n} e^{\beta\langle y_j, y_m\rangle}(y_m)_v \right) \Bigg] \\
&\quad \cdot \left( \sum_{l=1}^{n} e^{\beta\langle y_j, y_l\rangle} \right)^{-2}.
\end{aligned}
\tag{57}
$$

For $x, y \in \mathbb{R}^d$, we use $x \otimes y$ to denote the $d \times d$ matrix with $(u, v)$-th element $(x \otimes y)_{uv} = (x)_u(y)_v$, i.e., $x \otimes y := xy^T$. We then have the following proposition.

**Lemma B.3** *Adopt Assumption 2 and Equation (5). For any $i, j \in [\![1, n]\!]$, consider the $d \times d$ matrix formed by $\frac{\partial(\mathsf{ATT}(N(x_j)))_v}{\partial(x_i)_u}$, for $u, v \in [\![1, d]\!]$. Denote $y_k = N(x_k)$ for each $k \in [\![1, n]\!]$. Then, this matrix has the following form:*

$$\left( \frac{\partial(\mathsf{ATT}(N(x_j)))_v}{\partial(x_i)_u} \right)_{d \times d} = \|x_i\|^{-\frac{1}{2}} \left[ (\mathbf{R}_1 + \mathbf{R}_2)Z_j - (\mathbf{U}_1 + \mathbf{U}_2) \otimes \mathbf{V}_j \right] \cdot Z_j^{-2}, \qquad (58)$$

*where $Z_j = \sum_{l=1}^n e^{\beta\langle y_j, y_l\rangle}$ as in Equation (28),*

$$\mathbf{R}_1 := \delta_{ij}\beta \left( \mathbf{W}_j - y_i \otimes (\mathbf{W}_j y_i) \right), \quad \mathbf{R}_2 := e^{\beta\langle y_j, y_i\rangle} \left( (-y_i + \beta\mathbf{P}_{y_i}y_j) \otimes y_i + I_d \right), \qquad (59)$$

*and*

$$\mathbf{U}_1 := \delta_{ij}\beta \left( \mathbf{P}_{y_i}\mathbf{V}_j \right), \quad \mathbf{U}_2 := \beta e^{\beta\langle y_j, y_i\rangle} \left( \mathbf{P}_{y_i}y_j \right). \qquad (60)$$

*In Equation (59) and Equation (60),*

$$\mathbf{V}_j := \sum_{m=1}^n e^{\beta\langle y_j, y_m\rangle} y_m, \quad \mathbf{W}_j := \sum_{m=1}^n e^{\beta\langle y_j, y_m\rangle} y_m \otimes y_m, \quad \mathbf{P}_x y := y - \langle y, x\rangle x. \qquad (61)$$

**Proof 10 (Proof of Lemma B.3)** *By chain rule and Proposition B.1, we have that*

$$\begin{aligned}
\frac{\partial(\mathsf{ATT}(N(x_j)))_v}{\partial(x_i)_u} &= \sum_{k=1}^n \sum_{w=1}^d \frac{\partial(\mathsf{ATT}(y_j))_v}{\partial(y_k)_w}\bigg|_{Y=\mathcal{N}(X)} \cdot \frac{\partial(N(x_k))_w}{\partial(x_i)_u} \\
&= \|x_i\|^{-\frac{3}{2}} \left( \|x_i\| \cdot \frac{\partial(\mathsf{ATT}(y_j))_v}{\partial(y_i)_u} - \sum_{w=1}^d (x_i)_u(x_i)_w \frac{\partial(\mathsf{ATT}(y_j))_v}{\partial(y_i)_w} \right)\bigg|_{Y=\mathcal{N}(X)} \\
&= \|x_i\|^{-\frac{1}{2}} \left( \frac{\partial(\mathsf{ATT}(y_j))_v}{\partial(y_i)_u} - (y_i)_u \sum_{w=1}^d (y_i)_w \frac{\partial(\mathsf{ATT}(y_j))_v}{\partial(y_i)_w} \right)\bigg|_{Y=\mathcal{N}(X)}.
\end{aligned} \qquad (62)$$

*According to Proposition B.2 and the notation $Z_j = \sum_{l=1}^n e^{a_{jl}}$, we see that*

$$\begin{aligned}
\sum_{w=1}^d (y_i)_w &\frac{\partial(\mathsf{ATT}(y_j))_v}{\partial(y_i)_w} \\
&= \left[ \left( \delta_{ij}\beta \left( \sum_{m=1}^n e^{\beta\langle y_j, y_m\rangle} \langle y_m, y_i\rangle (y_m)_v \right) + e^{\beta\langle y_j, y_i\rangle} (\beta\langle y_j, y_i\rangle + 1)(y_i)_v \right) \cdot Z_j \right. \\
&\quad \left. - \left( \delta_{ij}\beta \left( \sum_{l=1}^n e^{\beta\langle y_j, y_l\rangle} \langle y_l, y_i\rangle \right) + \beta e^{\beta\langle y_j, y_i\rangle} \langle y_j, y_i\rangle \right) \cdot \left( \sum_{m=1}^n e^{\beta\langle y_j, y_m\rangle} (y_m)_v \right) \right] \cdot Z_j^{-2}.
\end{aligned} \qquad (63)$$

*Hence,*

$$\|x_i\|^{\frac{1}{2}} \cdot \frac{\partial(\mathsf{ATT}(N(x_j)))_v}{\partial(x_i)_u}$$

$$= \left( \frac{\partial(\mathsf{ATT}(y_j))_v}{\partial(y_i)_u} - (y_i)_u \sum_{w=1}^{d} (y_i)_w \frac{\partial(\mathsf{ATT}(y_j))_v}{\partial(y_i)_w} \right) \Bigg|_{Y=\mathcal{N}(X)}$$

$$= \left[ \left[ \delta_{ij}\beta \left( \sum_{m=1}^{n} e^{\beta\langle y_j, y_m\rangle} \left( (y_m)_u(y_m)_v - \langle y_m, y_i\rangle(y_m)_v(y_i)_u \right) \right) \right.\right.$$

$$\left. + e^{\beta\langle y_j, y_i\rangle} \left( \beta(y_j)_u(y_i)_v + \delta_{uv} - (\beta\langle y_j, y_i\rangle + 1)(y_i)_v(y_i)_u \right) \right] \cdot Z_j \tag{64}$$

$$- \left[ \delta_{ij}\beta \left( \sum_{l=1}^{n} e^{\beta\langle y_j, y_l\rangle} \left( (y_l)_u - \langle y_l, y_i\rangle(y_i)_u \right) \right) \right.$$

$$\left.\left. + \beta e^{\beta\langle y_j, y_i\rangle} \left( (y_j)_u - \langle y_j, y_i\rangle(y_i)_u \right) \right] \cdot \left( \sum_{m=1}^{n} e^{\beta\langle y_j, y_m\rangle}(y_m)_v \right) \right] \cdot Z_j^{-2}.$$

*We then adopt the notation Equation (61), i.e.,*

$$\mathbf{V}_j = \sum_{m=1}^{n} e^{\beta\langle y_j, y_m\rangle} y_m, \quad \mathbf{W}_j = \sum_{m=1}^{n} e^{\beta\langle y_j, y_m\rangle} y_m \otimes y_m, \quad \mathbf{P}_x y := y - \langle y, x\rangle x. \tag{65}$$

*So, the matrix form of Equation (64) becomes*

$$\left[ \left[ \delta_{ij}\beta(\mathbf{W}_j - y_i \otimes (\mathbf{W}_j y_i)) + e^{\beta\langle y_j, y_i\rangle}(\beta y_j \otimes y_i + I_d - (\beta\langle y_j, y_i\rangle + 1)y_i \otimes y_i) \right] \cdot Z_j \right.$$

$$- \left[ \delta_{ij}\beta(\mathbf{V}_j - \langle \mathbf{V}_j, y_i\rangle y_i) + \beta e^{\beta\langle y_j, y_i\rangle}(y_j - \langle y_j, y_i\rangle y_i) \right] \otimes \mathbf{V}_j \right] \cdot Z_j^{-2}$$

$$= \left[ \left[ \delta_{ij}\beta(\mathbf{W}_j - y_i \otimes (\mathbf{W}_j y_i)) + e^{\beta\langle y_j, y_i\rangle}((-y_i + \beta\mathbf{P}_{y_i}y_j) \otimes y_i + I_d) \right] \cdot Z_j \right. \tag{66}$$

$$- \left[ \delta_{ij}\beta(\mathbf{P}_{y_i}\mathbf{V}_j) + \beta e^{\beta\langle y_j, y_i\rangle}(\mathbf{P}_{y_i}y_j) \right] \otimes \mathbf{V}_j \right] \cdot Z_j^{-2}.$$

*We further use the notations in Equation (59) and Equation (60), i.e.,*

$$\mathbf{R}_1 = \delta_{ij}\beta e^{\beta\rho}(\mathbf{W}_j - y_i \otimes (\mathbf{W}_j y_i)), \quad \mathbf{R}_2 = e^{\beta\langle y_j, y_i\rangle}((-y_i + \beta\mathbf{P}_{y_i}y_j) \otimes y_i + I_d), \tag{67}$$

*and*

$$\mathbf{U}_1 = \delta_{ij}\beta(\mathbf{P}_{y_i}\mathbf{V}_j), \quad \mathbf{U}_2 = \beta e^{\beta\langle y_j, y_i\rangle}(\mathbf{P}_{y_i}y_j). \tag{68}$$

*Finally, the matrix form of Equation (64) becomes*

$$[(\mathbf{R}_1 + \mathbf{R}_2)Z - (\mathbf{U}_1 + \mathbf{U}_2) \otimes \mathbf{V}_j] \cdot Z_j^{-2}. \tag{69}$$

**Lemma B.4** *Let $\beta = \gamma \log n$ where $\gamma$ is a positive constant. Under Assumption 2 and Equation (5), if $\gamma > \frac{1}{1-\rho_2}$, then for any fixed $i, j \in [\![1, n]\!]$, the $d \times d$ matrix satisfies*

$$\left( \frac{\partial(\mathsf{ATT}(N(x_j)))_v}{\partial(x_i)_u} \right)_{d \times d} = \frac{\delta_{ij}}{\|x_i\|}(I_d - y_i \otimes y_i) + \mathbf{o}_n(1) + o_n(1) \cdot I_d, \tag{70}$$

*where the leading order term is exactly $\frac{\partial(N(x_j))_v}{\partial(x_i)_u}$. The term $\mathbf{o}_n(1)$ ($o_n(1)$, respectively) is a $d \times d$ matrix (constant, respectively) with matrix norm as defined in Equation (17) (value, respectively) going to 0 as $n \to +\infty$, with a speed independent of $i, j$ but only depending on $\gamma, \rho_2, q_1$.*

**Proof 11 (Proof of Lemma B.4)** *We frequently use this formula: for two vectors $V_1, V_2$, the matrix norm of $V_1 \otimes V_2$ as defined in Equation (17) is $\|V_1\|\|V_2\|$. When $\gamma > \frac{1}{1-\rho_2}$, $n^\gamma > n^{1+\gamma\rho_2}$, and*

we know from Lemma A.1 that $Z_j = (1 + o_n(1)) \cdot e^\beta$ for any $j \in [\![1, n]\!]$. Adopt the notations in Proposition B.3, we then show the following facts when $\gamma > \frac{1}{1-\rho_2}$:

$$\mathbf{R}_1 Z_j^{-1} = \mathbf{o}_n(1), \quad \mathbf{R}_2 Z_j^{-1} = \delta_{ij} \left(-y_i \otimes y_i + I_d\right) + \mathbf{o}_n(1) + o_n(1) \cdot I_d, \tag{71}$$

and

$$[(\mathbf{U}_1 + \mathbf{U}_2) \otimes \mathbf{V}_j] \cdot Z_j^{-2} = \mathbf{o}_n(1). \tag{72}$$

First, for $\mathbf{R}_1 Z_j^{-1}$, when $i \neq j$, we have that $\mathbf{R}_1 = 0$ by its definition. When $i = j$, $\mathbf{R}_1 = \beta \sum_{m=1}^n e^{\beta \langle y_i, y_m \rangle} \left(y_m \otimes y_m - \langle y_m, y_i \rangle y_i \otimes y_m\right)$ and we notice that the term when $m = i$ is 0. So, because $\|y_m \otimes y_m - \langle y_m, y_i \rangle y_i \otimes y_m\| \leq \|y_m\|^2 + \|y_m\|^2 \|y_i\|^2 = 2$, $e^\beta = n^\gamma$,

$$\|\mathbf{R}_1\| Z_j^{-1} \leq \beta(n-1) e^{\beta \rho_2} \cdot 2 Z_j^{-1} \leq 2\gamma \log(n) \cdot n^{\gamma \rho_2 + 1 - \gamma}(1 + o_n(1)), \tag{73}$$

which goes to 0 with a speed independent of $i, j$, because $\gamma \rho_2 + 1 - \gamma < 0$.

For $\mathbf{R}_2 Z_j^{-1}$, we notice that when $i \neq j$, $e^{\beta \langle y_i, y_j \rangle} Z_j^{-1} \leq e^{\beta(\rho_2 - 1)}(1 + o_n(1)) = n^{\gamma(\rho_2 - 1)}(1 + o_n(1))$, which goes to 0 with a speed independent of $i, j$. So, $\mathbf{R}_2 Z_j^{-1} = \mathbf{o}_n(1) + o_n(1) \cdot I_d$ when $i \neq j$. When $i = j$, $\mathbf{R}_2 = e^\beta \left(-y_i \otimes y_i + I_d\right)$, and so $\mathbf{R}_2 Z_j^{-1} = \left(-y_i \otimes y_i + I_d\right) + \mathbf{o}_n(1) + o_n(1) \cdot I_d$.

For $[(\mathbf{U}_1 + \mathbf{U}_2) \otimes \mathbf{V}_j] \cdot Z_j^{-2}$, we see that when $i \neq j$, $\mathbf{U}_1 = 0$, and so

$$\|(\mathbf{U}_1 + \mathbf{U}_2) \otimes \mathbf{V}_j\| \cdot Z_j^{-2} \leq Z_j^{-2} \beta \sum_{m=1}^n e^{\beta \langle y_j, y_m + y_i \rangle} \|y_m\| \|\mathbf{P}_{y_i} y_j\|$$
$$\leq Z_j^{-2} \beta e^{\beta(1+\rho_2)} n = \gamma \log(n) n^{\gamma(\rho_2 - 1) + 1}(1 + o_n(1)), \tag{74}$$

which goes to 0 with a speed independent of $i, j$ because $\gamma > \frac{1}{1-\rho_2}$. When $i = j$, $\mathbf{U}_2 = 0$, and so

$$\|(\mathbf{U}_1 + \mathbf{U}_2) \otimes \mathbf{V}_j\| \cdot Z_j^{-2} \leq Z_j^{-2} \beta \|\mathbf{P}_{y_i} \mathbf{V}_i\| \|\mathbf{V}_i\|$$
$$\leq Z_j^{-2} \beta \left(\sum_{m \neq i} e^{\beta \langle y_i, y_m \rangle} \|\mathbf{P}_{y_i} y_m\|\right) \cdot \left(e^\beta + \sum_{m \neq i} e^{\beta \langle y_i, y_m \rangle} \|\mathbf{P}_{y_i} y_m\|\right)$$
$$\leq Z_j^{-2} \beta \left(e^{\beta \rho_2} n\right) \cdot \left(e^\beta + e^{\beta \rho_2} n\right)$$
$$= \gamma \log(n) n^{\gamma(\rho_2 - 1) + 1}(1 + n^{\gamma(\rho_2 - 1) + 1})(1 + o_n(1)), \tag{75}$$

which goes to 0 with a speed independent of $i, j$ because $\gamma > \frac{1}{1-\rho_2}$. Hence, $[(\mathbf{U}_1 + \mathbf{U}_2) \otimes \mathbf{V}_j] \cdot Z_j^{-2} = \mathbf{o}_n(1)$.

**Lemma B.5** *Let $\beta = \gamma \log n$ where $\gamma$ is a positive constant. Under Assumption 2 and Equation (5), if $\gamma < \frac{1}{1-\rho_1}$, then for fixed $i, j \in [\![1, n]\!]$, the $d \times d$ matrix satisfies*

$$\left\| \left(\frac{\partial (\mathsf{ATT}(N(x_j)))_v}{\partial (x_i)_u}\right)_{d \times d} \right\| \leq \|x_i\|^{-\frac{1}{2}} \cdot \left(2\beta \delta_{ij} + (2\beta + \sqrt{d}) e^{a_{ij}} Z_j^{-1}\right). \tag{76}$$

**Proof 12 (Proof of Lemma B.5)** *According to Lemma A.1, when $\gamma < \frac{1}{1-\rho_1}$, $Z_j = (1 + o_n(1)) \cdot \left(\sum_{k \neq j} e^{a_{jk}}\right)$ for any $j \in [\![1, n]\!]$, and $Z_j \geq n^{\gamma \rho_1 + 1}(1 + o_n(1)) > n^\gamma(1 + o_n(1))$, because $\gamma \rho_1 + 1 > \gamma$. Adopt the notations in Proposition B.3, we then show the following facts when $\gamma < \frac{1}{1-\rho_1}$:*

$$\|\mathbf{R}_1\| Z_j^{-1} \leq \delta_{ij} \beta, \quad \|\mathbf{R}_2\| Z_j^{-1} \leq Z_j^{-1} e^{a_{ij}} \left(\beta + \sqrt{d - 1}\right), \tag{77}$$

and

$$\|(\mathbf{U}_1 + \mathbf{U}_2) \otimes \mathbf{V}_j\| \cdot Z_j^{-2} \leq \beta \left(\delta_{ij} + e^{a_{ij}} Z_j^{-1}\right). \tag{78}$$

First, for $\mathbf{R}_1 Z_j^{-1}$, when $i \neq j$, we have that $\mathbf{R}_1 = 0$ by its definition. When $i = j$, $\mathbf{R}_1 = \beta \sum_{m=1}^{n} e^{\beta \langle y_i, y_m \rangle} (y_m \otimes y_m - \langle y_m, y_i \rangle y_i \otimes y_m)$. So, because we have that $\|y_m \otimes y_m - \langle y_m, y_i \rangle y_i \otimes y_m\| = \|\mathbf{P}_{y_i} y_n \otimes y_m\| = \|\mathbf{P}_{y_i} y_n\| \|y_m\| \leq 1$,

$$\|\mathbf{R}_1\| Z_j^{-1} \leq \beta Z_j \cdot \cdot Z_j^{-1} = \beta. \tag{79}$$

For $\mathbf{R}_2 Z_j^{-1}$, because $\| - y_i \otimes y_i + I_d \| = \sqrt{d-1}$, we have that

$$\|\mathbf{R}_2\| Z_j^{-1} \leq Z_j^{-1} e^{a_{ij}} \left( \beta + \sqrt{d-1} \right). \tag{80}$$

For $[(\mathbf{U}_1 + \mathbf{U}_2) \otimes \mathbf{V}_j] \cdot Z_j^{-2}$, we see that $\|\mathbf{V}_j\| \leq \sum_{m=1}^{n} e^{\beta \langle y_j, y_m \rangle} = Z_j$. Also, $\|\mathbf{U}_1\| Z_j^{-1} \leq \delta_{ij} \beta \|\mathbf{V}_j\| Z_j^{-1} \leq \delta_{ij} \beta$, $\|\mathbf{U}_2\| Z_j^{-1} \leq \beta e^{a_{ij}} Z_j^{-1}$. Hence, we have that

$$\|(\mathbf{U}_1 + \mathbf{U}_2) \otimes \mathbf{V}_j\| \cdot Z_j^{-2} \leq \beta \left( \delta_{ij} + e^{a_{ij}} Z_j^{-1} \right). \tag{81}$$

**Proof 13 (Proof of Theorem 2.5)** *Theorem 2.5 follows directly from Lemma B.4 and Lemma B.5.*

### B.2 Proof of Theorem 2.4

The proof for Theorem 2.4 requires more delicate arguments. The part when $\gamma > \frac{1}{1-\rho}$ in Theorem 2.4 directly follows from Lemma B.4, so we only focus on the part when $\gamma \leq \frac{1}{1-\rho}$. We remark that when $\gamma < \frac{1}{1-\rho}$, our result is that $\frac{1}{nd}\|\nabla_X X'\|^2 = 0 + o_n(1)$, which is a better estimate than Equation (22) in Theorem 2.5.

We first have the following lemma which replaces Lemma B.3 when we adopt Assumption 1.

**Lemma B.6** *Adopt Assumption 1 and Equation (5). For any $i, j \in [\![1, n]\!]$, consider the $d \times d$ matrix formed by $\frac{\partial(\mathsf{ATT}(N(x_j)))_v}{\partial(x_i)_u}$, for $u, v \in [\![1, d]\!]$. Denote $y_k = N(x_k)$ for each $k \in [\![1, n]\!]$. Then, this matrix has the following form:*

$$\left( \frac{\partial(\mathsf{ATT}(N(x_j)))_v}{\partial(x_i)_u} \right)_{d \times d} = q^{-\frac{1}{2}} [(\mathbf{R}_1 + \mathbf{R}_2) Z - (\mathbf{U}_1 + \mathbf{U}_2) \otimes (\mathbf{U}_3 + \mathbf{U}_4)] \cdot Z^{-2}, \tag{82}$$

*where $Z = e^{\beta} + (n-1)e^{\beta\rho}$,*

$$\mathbf{R}_1 := \delta_{ij} \beta e^{\beta\rho} \left( \mathbf{W} - y_i \otimes (\mathbf{W} y_i) \right), \quad \mathbf{R}_2 := e^{\beta \langle y_j, y_i \rangle} \left( (-y_i + \beta \mathbf{P}_{y_i} y_j) \otimes y_i + I_d \right), \tag{83}$$

*and*

$$\begin{aligned} \mathbf{U}_1 := \delta_{ij} \beta e^{\beta\rho} \left( \mathbf{P}_{y_i} \mathbf{V} \right), \quad \mathbf{U}_2 := \beta e^{\beta \langle y_j, y_i \rangle} \left( \mathbf{P}_{y_i} y_j \right), \\ \mathbf{U}_3 := (e^{\beta} - e^{\beta\rho}) y_j, \quad \mathbf{U}_4 := e^{\beta\rho} \mathbf{V}. \end{aligned} \tag{84}$$

*In Equation (83) and Equation (84),*

$$\mathbf{V} := \sum_{m=1}^{n} y_m, \quad \mathbf{W} := \sum_{m=1}^{n} y_m \otimes y_m, \quad \mathbf{P}_x y := y - \langle y, x \rangle x. \tag{85}$$

**Proof 14 (Proof of Lemma B.6)** *We first apply Lemma B.3 to get Equation (58). After replacing $\langle y_j, y_m \rangle = \rho$ for $m \neq j$, we can obtain Equation (82). The only remark is that the term $\delta_{ij}(\mathbf{W}_j - y_i \otimes (\mathbf{W}_j y_i))$ in $\mathbf{R}_1$ of Equation (59) is nonzero when $i = j$. Then, when $i = j$, $\mathbf{W}_i - y_i \otimes (\mathbf{W}_i y_i) = \sum_{m=1}^{n} e^{\beta \langle y_i, y_m \rangle} (y_m \otimes y_m - y_i \otimes y_m \langle y_m, y_i \rangle)$. If $m = i$, the summand $(y_m \otimes y_m - y_i \otimes y_m \langle y_m, y_i \rangle)$ becomes $0$. Hence, $\mathbf{W}_i - y_i \otimes (\mathbf{W}_i y_i) = \sum_{m \neq i} e^{\beta \langle y_i, y_m \rangle} (y_m \otimes y_m - y_i \otimes y_m \langle y_m, y_i \rangle) = e^{\beta\rho} \sum_{m \neq i} (y_m \otimes y_m - y_i \otimes y_m \langle y_m, y_i \rangle) = e^{\beta\rho} \left( \mathbf{W} - y_i \otimes (\mathbf{W} y_i) \right)$.*

Next, to compute the matrix norm of Equation (82), we see that for any matrix $K$, its matrix norm square equals to $\mathrm{Tr}(K^T K)$. Hence, the matrix norm square of Equation (82) equals to

$$\begin{aligned} q^{-1} Z^{-4} \cdot \Big( & \mathrm{Tr} \left[ Z^2 (\mathbf{R}_1 + \mathbf{R}_2)^T (\mathbf{R}_1 + \mathbf{R}_2) \right] - 2Z (\mathbf{U}_1 + \mathbf{U}_2)^T (\mathbf{R}_1 + \mathbf{R}_2)(\mathbf{U}_3 + \mathbf{U}_4) \\ & + \|\mathbf{U}_1 + \mathbf{U}_2\|^2 \|\mathbf{U}_3 + \mathbf{U}_4\|^2 \Big). \end{aligned} \tag{86}$$

We then compute these terms separately, and sum them in $i, j$. We first have the following basic equalities for the notations $\mathbf{V}, \mathbf{W}$ in Equation (85).

**Lemma B.7** *For the notations in Equation (85), i.e.,*

$$\mathbf{V} := \sum_{m=1}^{n} y_m, \quad \mathbf{W} := \sum_{m=1}^{n} y_m \otimes y_m, \quad \mathbf{P}_x y := y - \langle y, x \rangle x, \tag{87}$$

*we have that*

$$\text{Tr}(\mathbf{W}^2) = \sum_{m,l} \langle y_m, y_l \rangle^2 = n(n\rho^2 + (1 - \rho^2)),$$
$$\tag{88}$$
$$\text{Tr}(\mathbf{W}) = n, \quad \text{Tr}(\mathbf{W} y_i y_i^T) = n\rho^2 + (1 - \rho^2), \quad \|\mathbf{P}_{y_i} y_j\|^2 = 1 - \rho^2.$$

*Also,*

$$\mathbf{W} y_i = \sum_{m=1}^{n} \langle y_m, y_i \rangle y_m = (1 - \rho) y_i + \rho \mathbf{V},$$

$$\langle \mathbf{V}, y_i \rangle = \sum_{m=1}^{n} \langle y_m, y_i \rangle = n\rho + (1 - \rho),$$
$$\tag{89}$$
$$\|\mathbf{V}\|^2 = \sum_{m,l} \langle y_m, y_l \rangle = n + \rho n(n-1) = n(n\rho + (1 - \rho)),$$

$$\|\mathbf{P}_{y_i} \mathbf{V}\|^2 = \|\mathbf{V}\|^2 - \langle \mathbf{V}, y_i \rangle^2 = (n-1)(n\rho + (1 - \rho))(1 - \rho),$$
$$\|\mathbf{W} y_i\|^2 = n^2 \rho^3 + 3n\rho^2(1 - \rho) + (1 + 2\rho)(1 - \rho)^2.$$

**Proof 15 (Proof of Lemma B.7)** *Direct Computations.*

**Lemma B.8** *For terms $\mathbf{R}_1, \mathbf{R}_2$ in Lemma B.6, we have that*

$$\sum_{i,j} \text{Tr} \left[ (\mathbf{R}_1 + \mathbf{R}_2)^T (\mathbf{R}_1 + \mathbf{R}_2) \right]$$
$$= \beta^2 e^{2\beta\rho} n \left[ n^2 \rho^2 (1 - \rho) + n(1 - \rho)(1 + \rho - 3\rho^2) - (1 + 2\rho)(1 - \rho)^2 \right] \tag{90}$$
$$+ \beta e^{\beta(\rho+1)} n(n-1)(1 - \rho^2)$$
$$+ e^{2\beta}(d - 1)n + e^{2\beta\rho} \left[ \beta^2 (1 - \rho^2) + d - 1 \right] n(n-1).$$

*As a corollary, when we pick $\beta = \gamma \log n$, we have the following phase transition limits as $n \to +\infty$:*

$$\frac{1}{nZ^2} \sum_{i,j} \text{Tr} \left[ (\mathbf{R}_1 + \mathbf{R}_2)^T (\mathbf{R}_1 + \mathbf{R}_2) \right] = \begin{cases} \beta^2 \rho^2 (1 - \rho) + o_n(1) & \text{if } \gamma < \frac{1}{1-\rho}, \\ \frac{d-1+\beta^2 \rho^2 (1-\rho)}{4} + o_n(1) & \text{if } \gamma = \frac{1}{1-\rho}, \\ d - 1 + o_n(1) & \text{if } \gamma > \frac{1}{1-\rho}. \end{cases} \tag{91}$$

**Proof 16 (Proof of Lemma B.8)** *We first notice that $\mathbf{W}$ is a symmetric matrix and $\|y_i\| = 1$. We then expand each term in Lemma B.8 and use Lemma B.7.*

$$\sum_{i,j} \text{Tr} \left[ (\mathbf{R}_1)^T \mathbf{R}_1 \right] = \beta^2 e^{2\beta\rho} \sum_i \left( \text{Tr} \left( \mathbf{W}^2 - 2\mathbf{W} y_i (\mathbf{W} y_i)^T \right) + \|y_i\|^2 \|\mathbf{W} y_i\|^2 \right)$$
$$= \beta^2 e^{2\beta\rho} \sum_i \left( \text{Tr} \mathbf{W}^2 - 2\|\mathbf{W} y_i\|^2 + \|\mathbf{W} y_i\|^2 \right) \tag{92}$$
$$= \beta^2 e^{2\beta\rho} n \left[ n^2 \rho^2 (1 - \rho) + n(1 - \rho)(1 + \rho - 3\rho^2) - (1 + 2\rho)(1 - \rho)^2 \right].$$

*Then,*

$$\sum_{i,j} \text{Tr} \left[ (\mathbf{R}_1)^T \mathbf{R}_2 \right] = \text{Tr} \sum_{i,j} \delta_{ij} \beta e^{\beta\rho} \left( \mathbf{W} - \mathbf{W} y_i y_i^T \right) e^{\beta \langle y_j, y_i \rangle} \left( (-y_i + \beta \mathbf{P}_{y_i} y_j) \otimes y_i + I_d \right)$$
$$= \beta e^{\beta(\rho+1)} \text{Tr} \sum_i \left( \mathbf{W} - \mathbf{W} y_i y_i^T \right) \left( -y_i y_i^T + I_d \right) = \beta e^{\beta(\rho+1)} \text{Tr} \sum_i \left( \mathbf{W} - \mathbf{W} y_i y_i^T \right)$$
$$= \beta e^{\beta(\rho+1)} n(n-1)(1 - \rho^2),$$
$$\tag{93}$$

*where the second equality is because* $\mathbf{P}_{y_i} y_i = 0$.

$$
\begin{aligned}
\sum_{i,j} \mathrm{Tr}\left[(\mathbf{R}_2)^T \mathbf{R}_2\right] &= \sum_{i,j} e^{2\beta\langle y_j, y_i\rangle} \mathrm{Tr}\left[(-y_i + \beta\mathbf{P}_{y_i}y_j)y_i^T + I_d\right]\left(y_i(-y_i + \beta\mathbf{P}_{y_i}y_j)^T + I_d\right] \\
&= \sum_{i \neq j} e^{2\beta\rho}\left[(1 + \beta^2(1 - \rho^2)) - 2 + d\right] + \sum_i e^{2\beta}(d - 1) \\
&= e^{2\beta}(d - 1)n + e^{2\beta\rho}\left[\beta^2(1 - \rho^2) + d - 1\right]n(n - 1).
\end{aligned}
\tag{94}
$$

*Next, we show the asymptotics Equation (91) as* $n \to +\infty$. *According to Lemma A.1, we have that*

$$
Z = \begin{cases} (1 + o_n(1)) \cdot ne^{\beta\rho} & \text{if } \gamma < \frac{1}{1-\rho}, \\ (1 + o_n(1)) \cdot e^{\beta} & \text{if } \gamma > \frac{1}{1-\rho}. \end{cases}
\tag{95}
$$

*That is, when* $\gamma < \frac{1}{1-\rho}$, *the leading order terms are those terms involving* $ne^{\beta\rho}$, *and all the remaining terms go to* $0$ *after dividing* $ne^{\beta\rho}$; *when* $\gamma > \frac{1}{1-\rho}$, *the leading order terms are those terms involving* $e^{\beta}$, *and all the remaining terms go to* $0$ *after dividing* $e^{\beta}$. *Hence, when* $\gamma < \frac{1}{1-\rho}$, *the leading order term in Equation (90) is the term* $\beta^2 e^{2\beta\rho}n^3\rho^2(1 - \rho)$; *when* $\gamma > \frac{1}{1-\rho}$, *the leading order term is* $e^{2\beta}(d - 1)n$. *This proves Equation (91).*

**Lemma B.9** *For terms* $\mathbf{R}_1, \mathbf{R}_2, \mathbf{U}_1, \mathbf{U}_2, \mathbf{U}_3, \mathbf{U}_4$ *in Lemma B.6, we have that*

$$
\begin{aligned}
\sum_{i,j} &(\mathbf{U}_1 + \mathbf{U}_2)^T(\mathbf{R}_1 + \mathbf{R}_2)(\mathbf{U}_3 + \mathbf{U}_4) \\
&= \rho\beta^2 e^{2\beta\rho}(e^{\beta} - e^{\beta\rho})n(n - 1)(n\rho + (1 - \rho))(1 - \rho) \\
&\quad + \beta^2 e^{3\beta\rho}n(n - 1)(n\rho + (1 - \rho))^2(1 - \rho) \\
&\quad + \beta e^{\beta(2\rho+1)}n(n - 1)(n\rho + (1 - \rho))(1 - \rho) \\
&\quad + \beta e^{2\beta\rho}(e^{\beta} - e^{\beta\rho})(\beta\rho + 1)n(n - 1)(1 - \rho^2) \\
&\quad + \beta e^{3\beta\rho}n(n - 1)(n\rho + (1 - \rho))(\beta(1 - \rho^2) + (1 - \rho)).
\end{aligned}
\tag{96}
$$

*As a corollary, when we pick* $\beta = \gamma \log n$, *we have the following phase transition limits as* $n \to +\infty$:

$$
\frac{1}{nZ^3}\sum_{i,j}(\mathbf{U}_1 + \mathbf{U}_2)^T(\mathbf{R}_1 + \mathbf{R}_2)(\mathbf{U}_3 + \mathbf{U}_4) = \begin{cases} \beta^2\rho^2(1 - \rho) + o_n(1) & \text{if } \gamma < \frac{1}{1-\rho}, \\ \frac{\beta^2\rho^2(1-\rho)}{4} + o_n(1) & \text{if } \gamma = \frac{1}{1-\rho}. \\ 0 + o_n(1) & \text{if } \gamma > \frac{1}{1-\rho}. \end{cases}
\tag{97}
$$

**Proof 17 (Proof of Lemma B.9)** *We expand each term in Lemma B.9 and also apply Lemma B.7 to each term. We first estimate terms involving* $\mathbf{U}_1$.

$$
\begin{aligned}
\sum_{i,j} \mathbf{U}_1^T \mathbf{R}_1 \mathbf{U}_3 &= \sum_i \beta^2 e^{2\beta\rho}(e^{\beta} - e^{\beta\rho})\left(\mathbf{P}_{y_i}\mathbf{V}\right)^T\left(\mathbf{W} - y_i \otimes (\mathbf{W}y_i)\right)y_i \\
&= \beta^2 e^{2\beta\rho}(e^{\beta} - e^{\beta\rho})\sum_i \left(\mathbf{P}_{y_i}\mathbf{V}\right)^T \mathbf{W}y_i = \rho\beta^2 e^{2\beta\rho}(e^{\beta} - e^{\beta\rho})\sum_i \left(\mathbf{P}_{y_i}\mathbf{V}\right)^T \mathbf{V} \\
&= \rho\beta^2 e^{2\beta\rho}(e^{\beta} - e^{\beta\rho})n(n - 1)(n\rho + (1 - \rho))(1 - \rho),
\end{aligned}
\tag{98}
$$

*where the second and the third equality is because* $\langle \mathbf{P}_{y_i}\mathbf{V}, y_i \rangle = 0$.

$$
\begin{aligned}
\sum_{i,j} \mathbf{U}_1^T \mathbf{R}_1 \mathbf{U}_4 &= \sum_i \beta^2 e^{3\beta\rho}\left(\mathbf{P}_{y_i}\mathbf{V}\right)^T\left(\mathbf{W} - y_i \otimes (\mathbf{W}y_i)\right)\mathbf{V} \\
&= \beta^2 e^{3\beta\rho}\sum_i \left(\mathbf{P}_{y_i}\mathbf{V}\right)^T \mathbf{W}\mathbf{V} = \beta^2 e^{3\beta\rho}(n\rho + (1 - \rho))\sum_i \|\mathbf{P}_{y_i}\mathbf{V}\|^2 \\
&= \beta^2 e^{3\beta\rho}n(n - 1)(n\rho + (1 - \rho))^2(1 - \rho),
\end{aligned}
\tag{99}
$$

*where the second equality is because* $\langle \mathbf{P}_{y_i}\mathbf{V}, y_i \rangle = 0$.

$$\sum_{i,j} \mathbf{U}_1^T \mathbf{R}_2 \mathbf{U}_3 = \sum_i \beta e^{\beta(\rho+1)}(e^\beta - e^{\beta\rho})\left(\mathbf{P}_{y_i}\mathbf{V}\right)^T\left((-y_i + \beta\mathbf{P}_{y_i}y_i)\otimes y_i + I_d\right)y_i = 0, \quad (100)$$

*where the second equality is because* $\langle \mathbf{P}_{y_i}\mathbf{V}, y_i \rangle = 0$ *and* $\mathbf{P}_{y_i}y_i = 0$.

$$\begin{aligned}
\sum_{i,j} \mathbf{U}_1^T \mathbf{R}_2 \mathbf{U}_4 &= \sum_i \beta e^{\beta(2\rho+1)}\left(\mathbf{P}_{y_i}\mathbf{V}\right)^T\left((y_i + \beta\mathbf{P}_{y_i}y_i)\otimes y_i + I_d\right)\mathbf{V} \\
&= \beta e^{\beta(2\rho+1)}\sum_i \|\mathbf{P}_{y_i}\mathbf{V}\|^2 = \beta e^{\beta(2\rho+1)}n(n-1)(n\rho+(1-\rho))(1-\rho),
\end{aligned} \quad (101)$$

*where the second equality is because* $\langle \mathbf{P}_{y_i}\mathbf{V}, y_i \rangle = 0$ *and* $\mathbf{P}_{y_i}y_i = 0$.

*Next, we estimate the terms involving* $\mathbf{U}_2$. *We first recall that* $\mathbf{U}_2 = \beta e^{\beta\langle y_j, y_i\rangle}\left(\mathbf{P}_{y_i}y_j\right)$. *Because* $\mathbf{P}_{y_i}y_j = 0$ *when* $i = j$, *we can just replace* $e^{\beta\langle y_j, y_i\rangle}$ *with* $e^{\beta\rho}$ *in* $\mathbf{U}_2$, *i,e,* $\mathbf{U}_2 = \beta e^{\beta\rho}\left(\mathbf{P}_{y_i}y_j\right)$. *Hence,*

$$\sum_{i,j} \mathbf{U}_2^T \mathbf{R}_1 \mathbf{U}_3 = 0, \quad \sum_{i,j} \mathbf{U}_2^T \mathbf{R}_1 \mathbf{U}_4 = 0, \quad (102)$$

*because* $\delta_{ij}(\mathbf{P}_{y_i}y_j) = 0$ *for any* $i, j$ *in* $\mathbf{U}_2^T \mathbf{R}_1$.

$$\begin{aligned}
\sum_{i,j} \mathbf{U}_2^T \mathbf{R}_2 \mathbf{U}_3 &= \sum_{i,j} \beta e^{\beta(\rho+\langle y_j, y_i\rangle)}(e^\beta - e^{\beta\rho})\left(\mathbf{P}_{y_i}y_j\right)^T\left((-y_i + \beta\mathbf{P}_{y_i}y_j)\otimes y_i + I_d\right)y_j \\
&= \beta e^{2\beta\rho}(e^\beta - e^{\beta\rho})\sum_{i\neq j}\left(\mathbf{P}_{y_i}y_j\right)^T\left((-y_i + \beta\mathbf{P}_{y_i}y_j)\rho + y_j\right) \\
&= \beta e^{2\beta\rho}(e^\beta - e^{\beta\rho})(\beta\rho+1)\sum_{i\neq j}\|\mathbf{P}_{y_i}y_j\|^2 \\
&= \beta e^{2\beta\rho}(e^\beta - e^{\beta\rho})(\beta\rho+1)n(n-1)(1-\rho^2).
\end{aligned} \quad (103)$$

*where the second equality is because* $\mathbf{P}_{y_i}y_j \neq 0$ *only when* $i \neq j$, *on which* $\langle y_j, y_i\rangle = \rho$, *and the third equality is because* $\langle \mathbf{P}_{y_i}y_j, y_i\rangle = 0$.

$$\begin{aligned}
\sum_{i,j} \mathbf{U}_2^T \mathbf{R}_2 \mathbf{U}_4 &= \sum_{i,j} \beta e^{\beta(2\rho+\langle y_j, y_i\rangle)}\left(\mathbf{P}_{y_i}y_j\right)^T\left((-y_i + \beta\mathbf{P}_{y_i}y_j)\otimes y_i + I_d\right)\mathbf{V} \\
&= \beta e^{3\beta\rho}\sum_{i\neq j}\left(\beta\|\mathbf{P}_{y_i}y_j\|^2(n\rho+(1-\rho)) + \left(\mathbf{P}_{y_i}y_j\right)^T\mathbf{V}\right) \\
&= \beta e^{3\beta\rho}\sum_{i\neq j}\left(\beta(1-\rho^2)(n\rho+(1-\rho)) + (1-\rho)(n\rho+(1-\rho))\right) \\
&= \beta e^{3\beta\rho}n(n-1)(n\rho+(1-\rho))(\beta(1-\rho^2)+(1-\rho)).
\end{aligned} \quad (104)$$

*where the second equality is because* $\mathbf{P}_{y_i}y_j \neq 0$ *only when* $i \neq j$, *on which* $\langle y_j, y_i\rangle = \rho$, *and the third equality is because* $\langle \mathbf{P}_{y_i}y_j, y_i\rangle = 0$.

*The proof for Equation (97) is similar to the proof for Equation (91) in Lemma B.8. Notice that when* $\gamma < \frac{1}{1-\rho}$, *we need to pick up terms involving* $ne^{\beta\rho}$, *and the leading order term in Equation (96) is the one in the second line of Equation (96), which is* $\beta^2 n^4 e^{3\beta\rho}\rho^2(1-\rho)$; *when* $\gamma > \frac{1}{1-\rho}$, *after diving* $nZ^3$, *all terms in Equation (96) are* $o_n(1)$ *terms.*

**Lemma B.10** *For terms* $\mathbf{U}_1, \mathbf{U}_2, \mathbf{U}_3, \mathbf{U}_4$ *in Lemma B.6, we have that*

$$\begin{aligned}
\sum_{i,j} &\|\mathbf{U}_1 + \mathbf{U}_2\|^2\|\mathbf{U}_3 + \mathbf{U}_4\|^2 \\
&= \beta^2 e^{2\beta\rho}n(n-1)(n\rho+2)(1-\rho) \\
&\quad \cdot \left[(e^\beta - e^{\beta\rho})^2 + 2e^{\beta\rho}(e^\beta - e^{\beta\rho})(n\rho+(1-\rho)) + e^{2\beta\rho}n(n\rho+(1-\rho))\right].
\end{aligned} \quad (105)$$

*As a corollary, when we pick $\beta = \gamma \log n$, we have the following phase transition limits as $n \to +\infty$:*

$$\frac{1}{nZ^4} \sum_{i,j} \|\mathbf{U}_1 + \mathbf{U}_2\|^2 \|\mathbf{U}_3 + \mathbf{U}_4\|^2 = \begin{cases} \beta^2 \rho^2 (1 - \rho) + o_n(1) & \text{if } \gamma < \frac{1}{1-\rho}, \\ \frac{\beta^2 \rho (1-\rho)(1+3\rho)}{16} + o_n(1) & \text{if } \gamma = \frac{1}{1-\rho}, \\ 0 + o_n(1) & \text{if } \gamma > \frac{1}{1-\rho}. \end{cases} \quad (106)$$

**Proof 18 (Proof of Lemma B.10)** *We notice that $\langle \mathbf{U}_1, \mathbf{U}_2 \rangle = 0$ because $\delta_{ij} \mathbf{P}_{y_i} y_j = 0$ for any $i, j$. So,*

$$\begin{aligned} \|\mathbf{U}_1 + \mathbf{U}_2\|^2 &= \delta_{ij} \beta^2 e^{2\beta\rho} \|\mathbf{P}_{y_i} \mathbf{V}\|^2 + \beta^2 e^{2\beta\langle y_j, y_i \rangle} \|\mathbf{P}_{y_i} y_j\|^2 \\ &= \delta_{ij} \beta^2 e^{2\beta\rho} (n-1)(n\rho + (1-\rho))(1-\rho) + (1 - \delta_{ij}) \beta^2 e^{2\beta\rho} (1 - \rho^2), \end{aligned} \quad (107)$$

*where the second equality is because $e^{2\beta\langle y_j, y_i \rangle} \|\mathbf{P}_{y_i} y_j\|^2 \neq 0$ only if $i \neq j$, on which $e^{2\beta\langle y_j, y_i \rangle} \|\mathbf{P}_{y_i} y_j\|^2 = e^{2\beta\rho}(1 - \rho^2)$.*

$$\begin{aligned} \|\mathbf{U}_3 + \mathbf{U}_4\|^2 &= (e^\beta - e^{\beta\rho})^2 + 2e^{\beta\rho}(e^\beta - e^{\beta\rho})\langle \mathbf{V}, y_j \rangle + e^{2\beta\rho} \|\mathbf{V}\|^2 \\ &= (e^\beta - e^{\beta\rho})^2 + 2e^{\beta\rho}(e^\beta - e^{\beta\rho})(n\rho + (1-\rho)) + e^{2\beta\rho} n(n\rho + (1-\rho)), \end{aligned} \quad (108)$$

*which is independent of $i, j$. Hence,*

$$\begin{aligned} &\sum_{i,j} \|\mathbf{U}_1 + \mathbf{U}_2\|^2 \|\mathbf{U}_3 + \mathbf{U}_4\|^2 \\ &= \left[ \beta^2 e^{2\beta\rho} n(n-1)(n\rho + (1-\rho))(1-\rho) + n(n-1)\beta^2 e^{2\beta\rho}(1 - \rho^2) \right] \|\mathbf{U}_3 + \mathbf{U}_4\|^2 \\ &= \beta^2 e^{2\beta\rho} n(n-1)(n\rho + 2)(1-\rho) \|\mathbf{U}_3 + \mathbf{U}_4\|^2 \\ &= \beta^2 e^{2\beta\rho} n(n-1)(n\rho + 2)(1-\rho) \\ &\quad \cdot \left[ (e^\beta - e^{\beta\rho})^2 + 2e^{\beta\rho}(e^\beta - e^{\beta\rho})(n\rho + (1-\rho)) + e^{2\beta\rho} n(n\rho + (1-\rho)) \right]. \end{aligned} \quad (109)$$

*The proof for Equation (106) is similar to the proof for Equation (91) in Lemma B.8. Notice that when $\gamma < \frac{1}{1-\rho}$, we need to pick up terms involving $ne^{\beta\rho}$, and the leading order term in Equation (105) is is $\beta^2 n^5 e^{4\beta\rho} \rho^2 (1 - \rho)$; when $\gamma > \frac{1}{1-\rho}$, after diving $nZ^4$, all terms in Equation (96) are $o_n(1)$ terms.*

**Proof 19 (Proof of Theorem 2.4)** *As we have mentioned at the beginning of Appendix B.2, we only need to focus the case when $\gamma \leq \frac{1}{1-\rho}$, which follows directly from Lemma B.8, Lemma B.9, and Lemma B.10. We notice that, in these three lemmas, the leading order terms are the same, $\beta^2 \rho^2 (1 - \rho)$, which cancels in Equation (86). Hence, when $\gamma < \frac{1}{1-\rho}$, $\frac{1}{nd} \|\nabla_X X'\|^2 = 0 + o_n(1)$. When $\gamma = \frac{1}{1-\rho}$, we also only need to use the corresponding cases in these three lemmas and combine them in Equation (86) to get the conclusion in Theorem 2.4. One remark is that under Assumption 1, we have that $n \leq d$ implicitly. So, when $\gamma = \frac{1}{1-\rho}$, terms in Equation (86) involving $\frac{\beta^2}{d} = \frac{\gamma^2 (\log n)^2}{d}$ also become $o_n(1)$.*

## C  MODIFIED ASSUMPTIONS WITH MORE MEDIAN PHASES

In this section, we modify Assumption 2, so that we can prove the existence of three different phases like Lemma A.1, Theorem 2.3, Theorem 2.5. We remark that we only showed the existence of two phases (two extrema) in Lemma A.1, Theorem 2.3, Theorem 2.5, but it doesn't mean under Assumption 2, there is no other transition phase between these two phases (two extrema). Under the following Assumption 3, we can show there are indeed at least three phases. Recall that for any $i \in [\![1, n]\!]$, we defined $y_i = N(x_i)$.

**Assumption 3**

- *For any $i \in [\![1, n]\!]$, $\|x_i\|^2 \in [q_1, q_2]$ for some positive constants $q_1 \leq q_2$.*

- *There is a $\tau \in (0,1]$, four positive constants $\rho_3, \rho_4, \kappa_3, \kappa_4$ with $\rho_3 \leq \rho_4$, $\kappa_3 \leq \kappa_4$, and $\rho_4 < 1$, such that for any $i \in [\![1,n]\!]$, if we define*

$$\mathcal{K}_i = \{m \neq i \mid \langle y_m, y_i \rangle \in [\rho_3, \rho_4]\}, \tag{110}$$

  *then we have that*

$$\kappa_3 \leq \frac{|\mathcal{K}_i|}{n^\tau} \leq \kappa_4. \tag{111}$$

- *For any $i \in [\![1,n]\!]$ and any $j \notin \mathcal{K}_i \cup \{i\}$, $\langle y_i, y_j \rangle \in [\rho_1, \rho_2]$ for some nonnegative constants $\rho_1, \rho_2$ satisfying $\rho_1 \leq \rho_2 < \rho_3 \leq \rho_4$.*

- *For technical reason, we further assume that $(1-\tau)(1-\rho_2) + \rho_2 < \rho_3$.*

**Lemma C.1** *Let $\beta = \gamma \log n$ where $\gamma$ is a positive constant. Under Assumption 2 and Equation (5), for any $i \in [\![1,n]\!]$,*

$$Z_i = \begin{cases} (1 + o_n(1)) \cdot \left( \sum_{m \notin \mathcal{K}_i \cup \{i\}} e^{a_{im}} \right) & \text{if } \gamma < \min\left\{\frac{1}{1-\rho_1}, \frac{1-\tau}{\rho_4-\rho_1}\right\}, \\ (1 + o_n(1)) \cdot \left( \sum_{m \in \mathcal{K}_i} e^{a_{im}} \right) & \text{if } \frac{1-\tau}{\rho_3-\rho_2} < \gamma < \frac{\tau}{1-\rho_3}, \\ (1 + o_n(1)) \cdot e^\beta & \text{if } \gamma > \max\left\{\frac{1}{1-\rho_2}, \frac{\tau}{1-\rho_4}\right\}, \end{cases} \tag{112}$$

*where the terms $o_n(1)$ go to 0 as $n \to +\infty$ with speeds independent of $i$ but only depending on $\gamma, \rho_1, \rho_2, \rho_3, \rho_4, \tau, \kappa_3, \kappa_4$.*

**Proof 20** *The proof is similar to Lemma A.1. We notice that*

$$\begin{aligned} Z_i &= e^\beta + \sum_{m \in \mathcal{K}_i} e^{a_{im}} + \sum_{m \notin \mathcal{K}_i \cup \{i\}} e^{a_{im}} \\ &= n^\gamma + \sum_{m \in \mathcal{K}_i} n^{\gamma \langle y_i, y_m \rangle} + \sum_{m \notin \mathcal{K}_i \cup \{i\}} n^{\gamma \langle y_i, y_m \rangle}. \end{aligned} \tag{113}$$

*We also notice that $\kappa_3 n^\tau \leq |\mathcal{K}_i| \leq \kappa_4 n^\tau$ according to Assumption 3. Hence,*

$$\kappa_3 n^{\tau + \gamma \rho_3} \leq |\mathcal{K}_i| \cdot n^{\gamma \rho_3} \leq \sum_{m \in \mathcal{K}_i} n^{\gamma \langle y_i, y_m \rangle} \leq |\mathcal{K}_i| \cdot n^{\gamma \rho_4} \leq \kappa_4 n^{\tau + \gamma \rho_4}, \tag{114}$$

*and*

$$(n - \kappa_4 n^\tau - 1) \cdot n^{\gamma \rho_1} \leq \sum_{m \notin \mathcal{K}_i \cup \{i\}} n^{\gamma \langle y_i, y_m \rangle} \leq (n - |\mathcal{K}_i| - 1) \cdot n^{\gamma \rho_2} \leq n^{1 + \gamma \rho_2}. \tag{115}$$

*When $\gamma < \min\left\{\frac{1}{1-\rho_1}, \frac{1-\tau}{\rho_4-\rho_1}\right\}$, the leading order term in $Z_i$ is $\sum_{m \notin \mathcal{K}_i \cup \{i\}} n^{\gamma \langle y_i, y_m \rangle}$; when $\frac{1-\tau}{\rho_3-\rho_2} < \gamma < \frac{\tau}{1-\rho_3}$, the leading order term in $Z_i$ is $\sum_{m \in \mathcal{K}_i} n^{\gamma \langle y_i, y_m \rangle}$; when $\gamma > \max\left\{\frac{1}{1-\rho_2}, \frac{\tau}{1-\rho_4}\right\}$, the leading order term in $Z_i$ is $n^\gamma$. We also remark that the last assumption in Assumption 3 is to ensure the existence of the middle phase, i.e., $\frac{1-\tau}{\rho_3-\rho_2} < \gamma < \frac{\tau}{1-\rho_3}$. This finishes the proof for Lemma C.1 by similar arguments as in Lemma A.1.*

A direct corollary of Lemma C.1 is the following theorem.

**Theorem C.2** *Under Assumption 2 and Equation (5) we have the following phase transition phenomena: let $\beta = \gamma \log n$ where $\gamma$ is a positive constant. For any $i \in [\![1,n]\!]$ the updating dynamics Equation (5) can be written as*

$$x_i' = \alpha x_i + \begin{cases} \frac{\sum_{m \notin \mathcal{K}_i \cup \{i\}} e^{a_{im}} y_m}{\sum_{m \notin \mathcal{K}_i \cup \{i\}} e^{a_{im}}} + \mathbf{o}_n(1) & \text{if } \gamma < \min\left\{\frac{1}{1-\rho_1}, \frac{1-\tau}{\rho_4-\rho_1}\right\}, \\ \frac{\sum_{m \in \mathcal{K}_i} e^{a_{im}} y_m}{\sum_{m \in \mathcal{K}_i} e^{a_{im}}} + \mathbf{o}_n(1) & \text{if } \frac{1-\tau}{\rho_3-\rho_2} < \gamma < \frac{\tau}{1-\rho_3}, \\ y_i + \mathbf{o}_n(1) & \text{if } \gamma > \max\left\{\frac{1}{1-\rho_2}, \frac{\tau}{1-\rho_4}\right\}, \end{cases} \tag{116}$$

*The terms $\mathbf{o}_n(1)$ represent vectors in $\mathbb{R}^d$ with norms going to 0 as $n \to +\infty$, with a speed independent of $i$ but only depending on $\gamma, \rho_1, \rho_2, \rho_3, \rho_4, \tau, \kappa_3, \kappa_4$.*

The proof of Theorem C.2 is similar to Lemma C.1 so we omit its proof.

# D    Analysis of the $\beta_n \asymp \sqrt{\log n}$ scaling for i.i.d. Gaussian scores

This appendix provides a short heuristic derivation of the $\beta_n \asymp \sqrt{\log n}$ scaling for softmax attention when the raw attention scores $a_1, \ldots, a_n$ are modeled as independent $\mathcal{N}(0,1)$ random variables. The purpose is to contrast the behavior of this Gaussian setting with the geometric setting analyzed in the main text, where pairwise score gaps remain $O(1)$ and the critical scale becomes $\beta_n \asymp \log n$.

Let $a_1, \ldots, a_n$ be i.i.d. $\mathcal{N}(0,1)$ and denote by

$$a_1^\downarrow \geq a_2^\downarrow \geq \cdots \geq a_n^\downarrow$$

their order statistics. Set

$$t_n := \Phi^{-1}\left(1 - \frac{1}{n}\right) \sim \sqrt{2\log n},$$

where $\Phi$ is the standard normal CDF. It is classical (see Theorem 2.1.1 in De Haan & Ferreira (2006)) that for any fixed $k \geq 2$,

$$\left(t_n(a_1^\downarrow - a_2^\downarrow), \ldots, t_n(a_1^\downarrow - a_k^\downarrow)\right) \;\Rightarrow\; (E_1, \; E_1 + E_2, \; \ldots, \; E_1 + \cdots + E_{k-1}), \tag{117}$$

where $(E_i)_{i \geq 1}$ are i.i.d. $\mathrm{Exp}(1)$ random variables. Thus the gap scale between neighboring scores in the Gaussian model is $1/\sqrt{\log n}$.

Define the softmax weights and the top weight

$$A_j(n) := \frac{\exp(\beta_n a_j)}{\sum_{k=1}^n \exp(\beta_n a_k)}, \qquad A_1^\downarrow(n) := \max_{1 \leq j \leq n} A_j(n).$$

Using Equation (117), one can write heuristically

$$A_1^\downarrow(n) \approx \frac{1}{1 + \sum_{k=2}^\infty \exp\left\{-(\beta_n/\sqrt{2\log n})\, S_{k-1}\right\}}, \qquad S_{k-1} := E_1 + \cdots + E_{k-1}. \tag{118}$$

where we have replaced the finite sum by an infinite series, which captures the leading asymptotics. The critical scaling $\sqrt{\log n}$ shows up in Equation (118). For example, in the supercritical regime $\beta_n \gg \sqrt{\log n}$, we have the following:

**Proposition D.1 (Supercritical regime)** *If*

$$\frac{\beta_n}{\sqrt{\log n}} \;\longrightarrow\; \infty,$$

*then $A_1^\downarrow(n) \to 1$ as $n \to \infty$. In other words, the attention weights concentrate on the top-scoring token.*

**Proof 21 (Sketch of proof)** *In Equation (118), we know the sum*

$$Z = \sum_{k=2}^\infty \exp\left\{-(\beta_n/\sqrt{2\log n})\, S_{k-1}\right\}$$

*is almost surely finite. Since $\beta_n/\sqrt{\log n} \to \infty$, every such term in $Z$ vanishes in probability. Hence the denominator in Equation (118) tends to 1, and $A_1^\downarrow(n) \to 1$ as $n \to \infty$.*

Proposition D.1 shows that the softmax enters a regime in which a single index captures asymptotically all attention mass as soon as $\beta_n \gg \sqrt{\log n}$. The critical scale is thus determined by the requirement

$$\beta_n(a_1^\downarrow - a_k^\downarrow) = O(k) \quad \Longleftrightarrow \quad \beta_n \asymp \sqrt{\log n}.$$

In the geometric models analyzed in the main text (simplex and almost-simplex assumptions), the gaps between the top pairwise inner products remain of order $O(1)$ as $n \to \infty$. Consequently, the balancing argument in the Introduction section yields the critical scaling $\beta_n \asymp \log n$. This stands in clear contrast to REM-type Gaussian models, where the top gaps shrink at the $1/\sqrt{\log n}$ scale and thus produce the critical regime $\beta_n \asymp \sqrt{\log n}$. This explains why REM-type Gaussian models lead to the $\sqrt{\log n}$ scale, whereas geometric models naturally produce the $\log n$ scale that aligns with many practical strategies to avoid mixing in long-context attention.

