# OpenReview forum: "Critical attention scaling in long-context transformers"
_ICLR.cc/2026/Conference — ICLR 2026 Poster_

### Official Review · Reviewer_W7uw · 2025-10-31

**Soundness:** 3
**Presentation:** 3
**Contribution:** 3
**Rating:** 6
**Confidence:** 3

**Summary:**

This paper attempts to justify the choice of attention scaling factor used in long-context scenario from the theoretical perspective. Specifically, the authors identified that with scaling choice $\beta_n=\gamma\log n$, the attention dynamics demonstrate a transition phase.  To delve into this question, the authors adopt a simplified self-attention model (without parameterisation) for theoretical analysis from both forward/backward perspective under two separate assumptions. The final numerical analysis justifies the derived theory.

**Strengths:**

This paper is well-motivated that the choice of $\beta_n$ scaling in Qwen, SSMax, SWAN-GPT ($\log n$) is not justified from the theoretical perspective. To address this question, the authors derive their theory on a simplified self-attention network, which is more tractable from the theoretical perspective. The authors consider both forward and backward two directions, which are validated through numerical studies. I believe the theoretical framework could be beneficial to the community.

**Weaknesses:**

Since the main paper is built on the theoretical analysis on an over-simplified self-attention network (without parameters), the generalisation of the conclusion to the realistic transformer model remains unknown. I have concerns about two main issues in the theoretical framework:

1. This paper still discusses self-attention under the scope of LLMs, which are assumed to adopt causal mask in the attention computation. However, this paper seems to completely ignore this point in the proof (from proof 1 that the denominator is independent of the choice of $i$). I am aware that the authors are trying to understand the attention scaling by simplification, but I am wondering whether the conclusion is still valid considering causal mask in the framework.

2. Another practical consideration in LLM is the existence of attention sink [1], which means the first token has large attention score while its value state is almost zero [2, 3]. Additionally, [3] claimed that this is a way to mitigate over-smoothing in long context scenarios. I am seeking for the authors' opinion towards this as this challenges some assumptions in this paper, e.g., the assumption on $a_{ii}=1$ and $a_{ij}=\rho$.


References:\
[1] Xiao et al. Efficient streaming language models with attention sinks. ICLR 2024.\
[2] Gu et al. When Attention Sink Emerges in Language Models: An Empirical View. ICLR 2025.\
[3] Barbero et al. Why Do LLMs Attend to the First Token? COLM 2025.

**Questions:**

See the weakness.

---

> ### Author Response · Authors · 2025-11-24
> **Thank you for your review**
>
> We are grateful to the reviewer for the careful read, the constructive feedback, and the positive assessment of our paper. Below we address both points and will incorporate the clarifications and extensions into the final version.
>
> -----------------------------------------------------------------
>
> ### Weaknesses
>
> - **W1:** **This paper still discusses self-attention under the scope of LLMs, which are assumed to adopt causal mask in the attention computation.** We thank the reviewer for raising this important point regarding causal masking in practical LLMs. Our analysis focuses on the non-causal self-attention operator because it enables a clean characterization of the phase transition and the $\log n$ scaling mechanism. In this simplified setting, the denominator is indeed independent of the choice of $i$, which allows us to isolate the core geometric and probabilistic effects driving mixing and sparsity. A full extension of our proofs to the causal case is certainly possible, though it requires suitable assumptions and becomes technically more involved, because the effective sequence length is now token-dependent and the interactions between tokens are no longer symmetric, as seen in recent works on the self-attention mechanism with causal mask [1,2]. We are currently working on this direction in follow-up work and expect the same phase transition and scaling behavior to persist under appropriate assumptions.
>
> - **W2:** **Another practical consideration in LLM is the existence of attention sink.** We thank the reviewer for raising this thoughtful question regarding attention sinks and their role in practical LLMs. Attention sinks indeed introduce an additional mechanism that injects a BOS token that helps stabilize attention dynamics. We view this mechanism is complementary to, rather than contradictory with, the scaling behavior we study. In fact, the attention sink can be viewed as introducing an additive stabilizing effect by absorbing a portion of the attention mass, while our work focuses on a multiplicative stabilization mechanism through $\beta$ scaling. These two mechanisms address rank-collapse from different angles, and both can co-exist in practice. When a sink token can be incorporated into our framework with minor modifications to the assumptions (e.g., treating the sink as a special token with fixed score/value structure), and we expect the same phase transition and $\log n$ scaling behavior to persist under this new settings. We will cite the references accordingly, and add a discussion in the revision to clarify how attention sinks relate to our setting.
>
> ----------------------------------------------------------------
>
> **Once again, we would like to thank you for taking the time to review our paper. We hope our answers and efforts are sufficient for you to consider raising your score and confidence in the correctness of our work. We remain open to further discussion.**
>
> ----------------------------------------------------------------
>
> [1] X. Wu, A. Ajorlou, Y. Wang, S. Jegelka, and A. Jadbabaie, On the role of attention masks and layernorm in transformers, 2024. arXiv:2405.18781.
>
> [2] N. Karagodin, Y. Polyanskiy, and P. Rigollet, Clustering in causal attention masking, 2024. arXiv:2411.04990.

---

> > ### Comment · Reviewer_W7uw · 2025-11-28
> >
> > Thank you for your response. I will maintain my score to support the acceptance of this paper.

---

### Official Review · Reviewer_mEek · 2025-11-01

**Soundness:** 4
**Presentation:** 3
**Contribution:** 2
**Rating:** 6
**Confidence:** 2

**Summary:**

The paper analyzes a long-context self-attention layer in which the attention scores are scaled by a factor that grows with the logarithm of the context length. In a simplified pre-normalization setting where the key, query, and value matrices are all equal to the identity, the authors prove that this scaling produces a clear phase transition as the context length increases. In the subcritical regime, when the scaling factor grows too slowly, attention weights become almost uniform across tokens, leading to strong contraction and rank collapse.
In the critical regime, when the scaling grows at just the right rate, attention focuses on a small but non-trivial subset of tokens, resulting in sparse and content-adaptive mixing. In the supercritical regime, when the scaling grows too quickly, attention effectively reduces to the identity and cross-token interactions vanish.

**Strengths:**

Although I am not too theoretically grounded myself, I very much liked this paper and I hope some of my comments/suggestions are useful to improve the paper. I have added some strong points below.

- The paper is mathematically strong and offers a well-grounded analysis of a practically important issue of rank collapse in long-context transformers.
- The key result is simple yet powerful, providing a clear and interpretable explanation of why logarithmic scaling works and what happens when the scaling constant is too small or too large.
- I particularly like that the analysis connects both forward and backward dynamics. The insight that vanishing gradients appear in the subcritical regime helps explain and even motivate architectural design choices such as residual connections and normalization layers in transformer blocks.
- The work successfully bridges theoretical and practical perspectives, with immediate implications for stable long-context training.

**Weaknesses:**

- As in many theoretical studies, the analysis operates in a simplified regime (pre-norm, identity key/query/value matrices, no positional encoding, no multi-head or causal masking). This is understandable but limits direct applicability.

- Experiments are synthetic and single-layer; there are no demonstrations on pretrained or small-scale real models. Even a lightweight validation could have been an interesting addition, though I recognize that this is beyond the paper’s main focus.

**Questions:**

- Do you view the phenomenon described here as primarily length-driven token uniformity (present even in a single layer)? How does this relate to depth-driven rank collapse? If logarithmic scaling is applied, is the length-driven component largely neutralized, leaving only depth-driven collapse? Are there diagnostics that could distinguish the two mechanisms in trained models?
- Your Theorems 2.3 and 2.4 show phase transitions in Jacobian norms, strongly reminiscent of the vanishing-gradient and over-smoothing analyses in graph neural networks (Arroyo et. al. 2025). Can your results be interpreted as the transformer analogue of these phenomena? Do you see conceptual parallels between the two settings?
- Recent work on attention sinks argues that attention sinks emerge as a way to prevent mixing. Do you agree with the interpretation that such sink tokens act as a compensatory mechanism to stabilize mixing when attention heads operate near the subcritical regime?

Arroyo, Álvaro, et al. "On vanishing gradients, over-smoothing, and over-squashing in GNNs: Bridging recurrent and graph learning." arXiv preprint arXiv:2502.10818 (2025).

Barbero, Federico, et al. "Why do LLMs attend to the first token?." arXiv preprint arXiv:2504.02732 (2025).

---

> ### Author Response · Authors · 2025-11-24
> **Thank you for your review**
>
> We are grateful to the reviewer for the careful read, the thoughtful questions, and the positive assessment of our paper. Below is a concise, point‑by‑point response to your questions.
> ### Weaknesses
> - **W1:** We thank the reviewer for the forward-looking suggestion. The analysis in our paper can be extended to the multi-head setting under suitable assumptions, albeit with more technical proofs. We chose to present our assumptions in a more streamlined and intuitive form in order to clearly highlight the key mechanisms. We are also actively considering extensions that incorporate positional encodings and causal masking in our ongoing work. We add a remark in Section 2 clarifying how our assumptions can be extended to more practical cases.
> - **W2:** The current paper is purely theoretical, and our aim is to study a simple model that explains the correct scaling factor $\log n$ used in practical LLMs. A careful, large-scale empirical study on the adaptive scaling factor in practice would be computationally intensive and is beyond the scope of the current paper as well as the computational resources available to our group. Nevertheless, we are actively exploring several adaptive schemes and will report numerical results in follow-up work.
>
> ### Questions
> - **Q1:** We thank the reviewer for raising this perspective on length-driven vs. depth-driven rank collapse; it is indeed an interesting question. The single-layer collapse studied in our paper is due to the self-attention layer, and length-driven rank collapse occurs when every token attends to each other equally. In real LLMs, there is no purely length-driven collapse because of residual connections; the combined effect of residuals and self-attention can be derived from our results. Yet, as the reviewer noted, as tokens are processed through layers they can still gradually develop depth-driven collapse. Since token statistics evolve across layers, we need adaptive scaling, with the constant $\gamma$ in $\beta_n = \gamma \log n$ chosen in a layer-dependent manner. Applying this adaptive scaling is expected to mitigate depth-driven rank collapse, as reported in many empirical works.  At the end of Section 2.2, we also briefly discussed the possibility of generalizing our results into multiple-layers cases. We do not have a definitive diagnostic that cleanly separates the two mechanisms in LLMs. It would be interesting to design a diagnostic which is adaptive to realistic settings, but it lies outside the scope of the present paper.
> - **Q2:** This is indeed an excellent connection! In the subcritical regime our attention operator behaves like a stochastic mixing kernel, whose repeated application contracts to a low‑dimensional subspace, i.e., an over‑smoothing effect analogous to message‑passing in GNN. Our Jacobian‑norm phase transition mirrors vanishing‑gradient thresholds in GNN. Despite this conceptual similarity, rank collapse in self‑attention is more involved. Intuitively, the mechanisms of GNNs can be regarded as a multiplicative system of matrices irrelevant to the real geometric positions of tokens, while the key mechanism in transformers is the dynamically interacting mechanism which is governed by the positions of tokens. We will cite [1] in the revised paper, and add a discussion in conclusion to make this bridge with GNNs notion explicit.
> - **Q3:** The attention sink mitigates mixing by introducing a BOS token that absorbs redundant attention mass. While certainly interesting, this differs from our approach, which mitigates mixing by rescaling the temperature. Both the attention sink model and our model maintain sparse, content-adaptive attention at large but finite context lengths. Although they share the same goal, our results show that changing the constant $\gamma$ in $\beta=\gamma\log n$ adjusts the interaction range of tokens that each token concentrates on, which is distinct from attention sinks. Our theoretical results do suggest that operating near the subcritical regime may lead to excessive mixing, and that introducing extra structure such as sink tokens could, in principle, counteract this tendency. However, we do not have empirical evidence that learned sink tokens universally serve as a compensatory mechanism, nor do we claim that combining the two mechanisms improves LLM performance in practice. We will include a paragraph in the conclusion of the paper and cite the references accordingly.
> ---
>
> **Once again, we would like to thank you for taking the time to review our paper. We hope our answers and efforts are sufficient for you to consider raising your score and confidence in the correctness of our work. We remain open to further discussion.**
>
> ---
>
> [1] A. Arroyo, A. Gravina, B. Gutteridge, F. Barbero, C. Gallicchio, X. Dong, M. Bronstein, and P. Vandergheynst, On vanishing gradients, over-smoothing, and over-squashing in gnns: Bridging recurrent and graph learning, 2025. arXiv:2502.10818.

---

### Official Review · Reviewer_bDMM · 2025-11-08

**Soundness:** 3
**Presentation:** 2
**Contribution:** 2
**Rating:** 4
**Confidence:** 3

**Summary:**

As language models have grown, context lengths have increased while signal propagation in attention layers has degraded; phenomena such as rank collapse have been observed. Empirically, prior work tunes the inverse temperature $ \beta $ using a $(\log n)^k $ scaling (with $k>0$ varying across papers). Under a strongly simplified model (all attention weights set to the identity), this paper analyzes tractably when a phase transition at $ \beta=\gamma\log n $ occurs and characterizes its connection to rank collapse.

**Strengths:**

* The angle‑parameter characterization is clear, and comparing to $1/(1-\rho)$ gives a natural condition for the phase transition.
* The numerical simulations align with the theoretical predictions and provide empirical support.

**Weaknesses:**

* **Insufficient precision in comparisons to prior work.** The introduction distinguishes the REM setting as “dense,” but that distinction applies at the $a_{ij}$ stage; at the $A_{ij}= \exp(\beta a_{ij})$ stage, the  behavior can become sparse, so the separation is not clearly justified. The discussion is thin on why REM suggests  $(\log n)^{1/2}$ while the present model yields $\log n$.
* **Strength of assumptions.** The score assumption,  exact one dominant entry, others equal (off‑diagonals fixed at  $\rho$ before normalization),  abstracts away near‑ties that can naturally occur in self-attention in real data: in language inputs the same token (e.g., a subject) can reasonably appear multiple times in one sequence.  This also stems from assuming the weights are identity mappings; In the REM case,  generally, the number of  near-tie entries can be more than one.
* **Gap to implementation.** Evidence is limited for how conclusions transfer to systems with learned $(K,Q,V)$. It remains unclear which architectural components would break the conclusions; in particular, the connection between the observed phase transition here and constants or regimes discussed in prior work (e.g., $C$ of $\beta=C \log n$ in Table 1) is weak.
* **Independence of numerical validation.** Experiments closely mirror the theoretical assumptions, limiting their independence as evidence for practical application. Stability of the critical behavior under slight deviations from the assumptions and reproducibility in more implementation‑like settings are underexplored. (e.g. incleasing the number of the near-tie entries)
* **Tone of claims.** While the $ \beta\asymp\log n $ phase‑transition picture is clear, the evidence is not yet sufficient to claim “clarifying  why logarithmic scaling maintains sparse, content-adaptive
attention at large context lengths.” for general practical attention scaling in my current understanding. The scope and limits of applicability should be stated more explicitly.

**Questions:**

* **Size of the top set.**  Because of $K=Q=I$, we get exactly one large  socre, others small scores after applying softmax with the Assumption 1.  How is Assumption 1 justified for self-attetoin with $K=Q=I$? Prior citations may support it, but a fresh statement of the authors’ view would help.   Any empirical evidence that appearance of exactly one large score in deep model (initialization/pretrain/finetuning phase) would also be helpful.

* **Consistency of $ \log n $ vs  $\sqrt{\log n}$.** Which specific assumption differences (e.g.\,correlation structure, sparsity/density/ layer norm) could determine the order of the critical scale?
* **Robustness to assumptions.** For non‑identity weights, to what extent do conclusions (including near‑ties) preserve the phase diagram and the critical behavior? Under the stated constraints, in what sense does the work clarify the claim that “logarithmic scaling maintains sparse, content‑adaptive attention at large context lengths”? Also, how do the authors explain $ (\log n)^2 $ (YaRN) scalings?

---

> ### Author Response · Authors · 2025-11-24
> **Thank you for your review (part 1)**
>
> We thank you for your detailed and thoughtful review. Your comments helped us sharpen both the technical details and the scope of our claims. Below we address each weakness and question in turn and describe how we will revise the paper.
>
> --------------------------------------------------------------
>
> ### Weaknesses
>
> - **W1:** **Insufficient precision in comparisons to prior work.** You are right that our comparison to the REM-based analysis of Giorlandino-Goldt was too terse. The key difference between our setting and Giorlandino-Goldt's is *not* sparsity, but the underlying geometric model of the scores. In the REM setting, the attention scores $a_{ij}$ are modeled essentially as (correlated) Gaussian "energies," and one can actually see the same $\sqrt{\log n}$ scaling in the simpler case where the $a_{ij}$ are i.i.d. normal. Clearly such Gaussian model carries no geometric information about the token embeddings. The reasoning behind the scaling $\beta_n = O(\sqrt{\log n})$ is purely due to the fact that the gap between the largest and second-largest entries in $\lbrace a_{ij}\rbrace_{j=1}^n$ is of order $1/\sqrt{\log n}$, so choosing $\beta_n \gg \sqrt{\log n}$ forces the softmax to concentrate almost all mass on a single largest entry. In that framework, a choice $\beta_n \sim \log n$ therefore places the system deep in the supercritical regime for large $n$.
>
>   To see why a $\sqrt{\log n}$ scale appears for Gaussian scores, let $a_1,\dots,a_n$ be i.i.d. $\mathcal{N}(0,1)$ and write the order statistics as $a_1^\downarrow \ge \cdots \ge a_n^\downarrow$. It is a standard result that $a_1^\downarrow-a_k^\downarrow = \Theta(k/\sqrt{\log n})$.
>
>   Now consider the top softmax weight at $\beta_n$:
>   $$
>   A_1^\downarrow  =  \frac{1}{1+\sum_{k=2}^n \exp\{-\beta_n(a_1^\downarrow-a_k^\downarrow)\}}.
>   $$
>   Using the estimate above,
>   $$
>   A_1^\downarrow  \approx  \frac{1}{1+\sum_{k=2}^\infty \exp\{- (Ck\beta_n/\sqrt{\log n}) \}},
>   $$
>   If $\beta_n/\sqrt{\log n}\to \infty$, each exponential term vanishes and $A_1^\downarrow \to 1$.
>   This justifies the $\sqrt{\log n}$ scaling in the Gaussian (REM-like) case.
>
>   In contrast, our model explicitly imposes a simple geometric structure on the token representations (simplex / almost-simplex assumptions), which retains $\Theta(1)$ gaps between ordered scores (rather than $\sqrt{\log n}$) and yields the critical scaling $\beta_n \asymp \log n$ that is observed in many practical long-context methods. To make this distinction precise, we will add to the appendix of the revised paper a short heuristic proof of the supercritical regime of the i.i.d. Gaussian score model, and we will expand the introduction to explain how this contrasts with the geometric setting we analyze.

---

> ### Author Response · Authors · 2025-11-24
> **Thank you for your review (part 2)**
>
> - **W2:** **Strength of assumptions.** We understand the reviewer's concern that Assumption 1 is strong on its own, but we would like to note that we have relaxed it in two ways in the paper. First, the almost-simplex Assumption 2 allows perturbations of the ideal simplex geometry. Second, Assumption 3 in Appendix C further admits a non-trivial intermediate regime that allows for ties. What is important for our purposes is that the *scaling* remains the same under any of the three assumptions: the critical order $\beta_n \asymp \log n$ does not change. Assumptions 2 and 3 are therefore meant to show that the idealized Assumption 1 is already sufficient to predict the correct scaling behavior in more realistic configurations. Moreover, under Assumptions 2 and 3, it is straightforward to generalize the arguments to allow repeated tokens, and the same critical scaling $\beta_n \asymp \log n$ continues to hold. We will add a remark in Section 2.2 clarifying how the analysis adapts in the presence of repeated tokens.
>
> - **W3:** **Gap to implementation.** This is, as with all theoretical work, a fair criticism, but we actually choose our theoretical setup intentionally: our aim is not an exhaustive empirical study of $\beta_n$ across various architectures, but to establish the order-of-magnitude law and elicit the mechanism that establishes $\log n$ as the *critical* scale separating the diffusive and single-winner regimes predicted under Assumptions 1, 2 and 3.
>   In particular, the analyses under Assumptions 2 and 3 are already closer to realistic scenarios, and both exhibit a phase transition at order $\log n$. These assumptions are not intended to capture every possible case encountered in practice; instead, they retain a degree of generality while still enabling a tractable and interpretable analysis.
>
> - **W4:** **Independence of numerical validation.** We agree that the experiments are closely aligned with our theoretical setup, but they are not a simple restatement of the theorems. In Figures 1 and 2 the tokens are sampled at random, so each attention row typically exhibits multiple near-ties rather than a single dominant entry. The resulting phase diagrams still display the same critical transition at a $\log n$ scale between diffuse and single winner regimes, which indicates robustness to deviations from Assumptions 1 and 2. We also reiterate that Assumptions 2 and 3 are specifically designed to explore the critical behavior under deviations from the idealized Assumption 1. As noted in our response to the second weakness point, the analysis in the paper can be extended to settings with multiple near-tie entries. Our goal in this paper is to propose and analyze a simple model for long-context self-attention, and a comprehensive evaluation on full LLMs would be valuable but far beyond our present scope.
>
> - **W5:** **Tone of claims.** While we appreciate the reviewer's pushback on our wording, we respectfully maintain our original position. A central theme of the paper is precisely to clarify why logarithmic scaling preserves sparse, content-adaptive attention at large but finite context lengths. By adjusting the parameter $\gamma$ in $\beta = \gamma \log n$, the attention mechanism can change the attention range of each token, and hence the dynamical behaviors of tokens fall in different phase transition regions corresponding to different $\gamma$. Hence, the statements of the theorems together with their proofs demonstrate that logarithmic scaling maintains content-adaptive attention at large context lengths $n$. Assumptions 2 and 3 were introduced to demonstrate that this phenomenon persists beyond the idealized simplex geometry and in settings that more closely resemble practical configurations. For these reasons, we believe the statement in the Introduction is sufficiently well-supported and extends to more general scenarios.

---

> ### Author Response · Authors · 2025-11-24
> **Thank you for your review (part 3)**
>
> ----------------------------------
>
> ### Questions
>
> - **Q1:** **Size of the top set.** It is reasonable for the reviewer to question the "single largest score" assumption. We emphasize that this assumption is not essential: the same phase transition with critical scaling $\beta_n \asymp \log n$ persists when the top attention score has ties or when there are multiple near-ties. As noted in our responses to the reviewer's W2, this robustness is captured by Assumptions 2 and 3, and it can also be seen in Figures 1--2, where randomized tokens create many near-ties yet exhibit the same critical behavior. We will add a remark in Section 2 clarifying how our conclusions adapt in the presence of ties/near-ties.
>
> - **Q2:** **Consistency of $\log n$ vs $\sqrt{\log n}$.** We thank the reviewer for this insightful question. As noted in our reply to the reviewer's W1, the key distinction between our model and the REM framework lies in the behavior of the ordered scores $a_{ij}$: in REM, the gaps between ordered scores scale with $1/\sqrt{\log n}$, whereas in our geometric model the gaps remain $O(1)$. This difference reflects the combined influence of architecture and the distribution/geometry of the input tokens. The present paper is intended to identify a simple setting in which the $\log n$ scaling is critical and to highlight the associated phase transition. A more in-depth analysis of alternative scalings under different assumptions is ongoing work of the authors. We believe that a thorough treatment would merit a separate, full-length paper. As noted in our earlier response, we will also add to the appendix a short heuristic argument for the supercritical regime of the Gaussian score model, whose critical scaling is $\beta_n=\sqrt{\log n}$.
>
> - **Q3:** **Robustness to assumptions.** We appreciate the question. The aim of this paper is not an exhaustive, architecture-by-architecture study of $\beta_n$, but to analyze a simple mechanistic model that establishes the $\log n$ scaling widely used in practice. Although Assumption 1 is intentionally simple, we extend its applicability both theoretically (Assumptions 2 and 3, which admit almost-simplex geometry and near-ties) and empirically (Figures 1--2 with randomized tokens). By adjusting the parameter $\gamma$ in $\beta = \gamma \log n$, the attention mechanism can change the attention range of each token, and hence the dynamical behaviors of tokens fall in different phase transition regions corresponding to different $\gamma$. Taken together, these support our position that logarithmic scaling maintains sparse, content-adaptive attention at large context lengths for the standard transformer architecture used in many LLMs. We do not yet have a definitive explanation for the $(\log n)^2$ schedule reported in YaRN; our view is that it likely reflects an empirical fit tied to the specific positional-extrapolation scheme employed. We see this as a natural direction for follow-up work, and we would welcome further discussion on examining alternative (standard and non-standard) architectural choices beyond the setting analyzed here.
>
> -----------------------------------------------------------
>
> **Once again, we would like to thank you for taking the time to review our paper. We hope our answers and efforts are sufficient for you to consider raising your score and confidence in the correctness of our work. We remain open to further discussion.**
>
> -----------------------------------------------------------
>
> [1] L. De Haan and A. Ferreira, Extreme value theory: an introduction, Springer, 2006.

---

> ### Author Response · Authors · 2025-12-02
> **Global Rebuttal Summary**
>
> When the OpenReview leak was announced, only Reviewer W7uw had reviewed our rebuttal and expressed satisfaction with our revisions. Because further discussions are now frozen, we present this concise summary of the changes made during the rebuttal phase.
>
> # Overview of the paper
>
> This paper develops a framework for understanding phase transitions in self-attention as the context length $n$ grows, identifying a critical scaling $\beta_n \asymp \log n$ that separates a subcritical contractive regime from a supercritical unchanged regime. A central message is that this transition is rooted in the geometry of the score landscape: in our model, the gaps between the top ordered scores remain $O(1)$, which leads to the $\log n$ scaling. We show that this scaling is robust under various perturbations and in settings permitting multiple near-ties, demonstrating that the $\log n$ law is a structural consequence of content-adaptive interactions.
>
> # Summary of Key Improvements in the Rebuttal Revision
>
> ### **Added a new appendix explaining the difference between the $\log n$ scaling in this paper and the $\sqrt{\log n}$ scaling in prior work**
>
> In Appendix D, we show a short heuristic derivation of the $\beta_n \asymp \sqrt{\log n}$ scaling for softmax attention when the raw attention scores $a_{1},\dots,a_{n}$ are modeled as independent $\mathcal{N}(0,1)$ random variables.
>
> The purpose is to contrast the behavior of this Gaussian setting with the geometric setting analyzed in the main text, where pairwise score gaps remain $O(1)$ and the critical scale becomes $\beta_n \asymp \log n$.
>
> ### **Clarified the adaption of theoretical analysis in this paper to more practical transformer models**
>
> Based on reviewers' comments on Assumption 1, we emphasized that Assumption 2 and 3 allow perturbations of the ideal simplex geometry in Assumption 1. The *scaling* remains the same in all three assumptions: the critical order $\beta_n \asymp \log n$ does not change. Assumptions 2 and 3 are therefore meant to show that the idealized Assumption 1 is already sufficient to predict the correct scaling behavior in more realistic configurations. Moreover, we added Remark 2.2 to discuss the generalization of our theoretical analysis to allow multiple exact ties and near ties in the top scores, where the same critical scaling $\beta_n \asymp \log n$ continues to hold. In our replies to the reviewers, we also clarified that our theoretical analysis works for multi-head transformer models, and for positions of tokens concentrating on several clusters.
>
> ###  **Expanded the comparison with the over-smoothing in graph neural networks (GNNs) and the attention sink in transformer models**
>
> Based on reviewers' comments on potential connections between the phase transition phenomena in this paper and the over-smoothing in GNNs and the attention sink in transformer models, we added Section 4 to compare this paper with over-smoothing and attention sinks. Specifically, the over-smoothing in GNNs is *depth-driven*: the network dynamics can be viewed as a multiplicative composition of fixed matrices, largely insensitive to the geometric arrangement of the input features. In contrast, self-attention dynamics exhibit *length-driven* collapse, where the rank collapse can occur within a single layer and depends strongly on token positions and the context length. Second, attention sinks introduce additive stabilizing tokens to prevent excessive mixing, whereas our results focus on a multiplicative mechanism with temperature rescaling. Both approaches aim to maintain meaningful attention patterns at long but finite context lengths, and the works in attention sinks address rank collapse from distinct and potentially complementary angles.
>
> # Final Remarks
>
> Reviewers' comments greatly helped authors improve the clarity and quality of the paper. We add Remark 2.2 under our Assumption 2 to clarify the robustness of our assumptions and the possible adaption of the theoretical analysis in this paper to more practical transformer models. The revised manuscript further incorporates a new appendix delineating the distinctions between the $\log n$ scaling in our work and the $\sqrt{\log n}$ scaling in prior literature, alongside a dedicated section exploring links to over-smoothing in GNNs and attention sinks in transformers.
>
> We hope this summary helps the new Area Chair in evaluating the updated submission, and we extend our sincere thanks to all reviewers for their insightful feedback.

---

### Official Review · Reviewer_UoUF · 2025-11-09

**Soundness:** 4
**Presentation:** 4
**Contribution:** 4
**Rating:** 8
**Confidence:** 3

**Summary:**

The paper presents a rigorous theoretical analysis of attention scaling in long-context transformers. By studying a simplified but informative self-attention model, the authors show that the system undergoes a sharp phase transition determined by a critical scaling factor $\beta$ proportional to $log(n)$. This result provides a clear theoretical explanation for the empirical success of scaling strategies used in recent models such as YaRN, SSMax, and SWAN-GPT. The analysis addresses both the contractive properties of the forward pass and gradient behavior in the backward pass, demonstrating that this critical scaling is necessary to preserve meaningful token interactions and maintain stable training dynamics.

**Strengths:**

The main strength of the paper lies in the quality of its theoretical analysis. The results are rigorous and presented in a way that remains accessible. The work provides a direct and mathematically grounded explanation for a technique currently used in state-of-the-art large language models, which is a valuable and rare contribution. Moreover, by isolating the mechanisms that lead to performance degradation in long-context tasks, the paper offers a framework that can guide the development of more targeted scaling strategies.

**Weaknesses:**

The model necessarily incorporates simplifying assumptions, and the experiments are correspondingly limited. However, given that the goal of the work is to isolate and explain the underlying scaling phenomenon, already well studied by practitioners, rather than to propose a new architecture or strategy, this scope seems appropriate. The limitations are acknowledged and do not undermine the core contribution.

**Questions:**

- The phase transition boundary depends on geometric quantities such as the inner product ρ and the norm q. The simplex case makes the nature of this dependence particularly transparent. Recent work has shown that layer normalization can systematically influence token norms over depth (e.g., “Normalization in Attention Dynamics,” arXiv:2510.22026). This raises the question of whether an adaptive scaling factor, adjusting $\beta$ based on observed per-layer statistics of token norms or pairwise similarities, could maintain the model closer to the critical regime and potentially improve performance?
- Assumption 2 requires a positive lower bound $\rho_1 > 0$ on pairwise inner products. While the upper bound $\rho_2 < 1$ is intuitive, the necessity of a positive lower bound is less obvious. Is this bound essential for the phase transition to appear, or is it primarily a technical condition to simplify the proof? Additional intuition here would be helpful.
- The analysis is conducted under the simplifying assumption $Q = K = I$, which is a standard and effective choice for achieving tractability and is consistent with related theoretical work. As a curiosity, do you observe the same phase transition behavior in numerical experiments when $Q$ and $K$ are non-identity matrices, for example in the case where $Q^TK$ is symmetric positive definite? If so, does the location of the phase transition or the strength of contraction change in a predictable way?

---

> ### Author Response · Authors · 2025-11-24
> **Thank you for your review**
>
> We are grateful to the reviewer for the careful read, the thoughtful questions, and the positive
> assessment of our paper. Below is a concise, point-by-point response to your questions.
>
> ---------------------------------------------------------
>
> ### Questions
>
> - **Q1:** **The phase transition boundary depends on geometric quantities such as the inner product $\rho$ and the norm $q$.** We thank the reviewer for the references and the forward-looking suggestion.
>   As the reviewer points out, our analysis shows that the phase boundary is controlled by the product of $\beta$ and layer-wise geometric statistics (token norms and pairwise similarities).
>   This naturally motivates *adaptive* inverse-temperature schedules and aligns with reports from recent LLMs.
>   While the reviewer raises a valuable question to study, a careful, large-scale empirical study on the adaptive scaling factor is beyond the scope of the current paper. Nevertheless, we are actively exploring several adaptive schemes and will report numerical results in follow-up work. Note that although this paper is presented using the pre-layer normalization in assumptions and theorems, post-layer normalization also exhibits phase transitions under the $\log n$-scaling, where the proofs and statements are similar. We also believe that similar scaling orders are universal for different layer normalizations since these can be studied under a unified framework as shown in [1].
>
> - **Q2:** **Assumption 2 requires a positive lower bound $\rho_1>0$ on pairwise inner products.** We thank the reviewer for the careful reading of Assumption 2. The condition $\rho_1 > 0$ is imposed for a technical reason: it guarantees the existence of a well-defined subcritical regime. On the other hand, because $A_{ij} = \exp{(\beta a_{ij})}$ and $\beta = \gamma \log n$ is a relatively large positive number, those weights with $a_{ij}<0$ have negligible effect compared to those weights with $a_{ij}>0$. For example, when there are two clusters of points of numbers of order $\sim n$ around the north pole and the south pole of the unit sphere in $\mathbb{R}^d$ respectively, arguments analogous to those in the paper can be carried out within each cluster, and phase transitions at order $\log n$ still arise. But formally extending the proof to this setting requires additional refinements that would distract from the main line of analysis. Our assumptions and techniques do not attempt to include all possible cases while still keep assumptions and arguments as general as possible.
>
> - **Q3:** **The analysis is conducted under the simplifying assumption $Q=K=I$.** Thank you for catching this point. Although we assume $Q = K = I$ for clarity, this choice is not essential to our analysis. An extension of the proofs to general query and key matrices $Q, K$ is certainly possible, though it requires additional assumptions and becomes technically more involved, because the token interactions are no longer symmetric. To illustrate, consider the general case with $a_{ij} = q_i^\top k_j$, $q_i = Q x_i$, $k_j = K x_j$, $M := Q^\top K$. Define the gap
>   $$
>   \Delta_i := \max_j a_{ij}  -  2\mathrm{n}\mathrm{d} \mathrm{m} \mathrm{a} \mathrm{x_j}  a_{ij},
>   $$
>   and let $J_\ast(i) = \arg\max_{j\in[n]} a_{ij}$ denote the set of maximizers. Following the same computation as in our paper, one obtains that when
>   $$
>   \beta_n > \frac{1}{\Delta_i} \log n,
>   $$
>   the $i$-th token attends only to the tokens in $J_\ast(i)$ in the limit, which corresponds to the supercritical regime. A further characterization of rank collapse even in this simple case is involved because the structure of maximizers is no longer symmetric. We therefore choose to focus on the current settings to clearly convey the core mechanisms.
>
> ---
>
> **Once again, we would like to thank you for taking the time to review our paper. We hope our answers have reinforced your positive evaluation of our work. We remain open to further discussion.**
>
> ---
>
> [1] N. Karagodin, S. Ge, Y. Polyanskiy, and P. Rigollet, Normalization in attention dynamics, 2025. arXiv:2510.22026.

---

### Author Response · Authors · 2025-12-02
**Global Rebuttal Summary**

When the OpenReview leak was announced, only Reviewer W7uw had reviewed our rebuttal and expressed satisfaction with our revisions. Because further discussions are now frozen, we present this concise summary of the changes made during the rebuttal phase.

# Overview of the paper

This paper develops a framework for understanding phase transitions in self-attention as the context length $n$ grows, identifying a critical scaling $\beta_n \asymp \log n$ that separates a subcritical contractive regime from a supercritical unchanged regime. A central message is that this transition is rooted in the geometry of the score landscape: in our model, the gaps between the top ordered scores remain $O(1)$, which leads to the $\log n$ scaling. We show that this scaling is robust under various perturbations and in settings permitting multiple near-ties, demonstrating that the $\log n$ law is a structural consequence of content-adaptive interactions.

# Summary of Key Improvements in the Rebuttal Revision

### **Added a new appendix explaining the difference between the $\log n$ scaling in this paper and the $\sqrt{\log n}$ scaling in prior work**

In Appendix D, we show a short heuristic derivation of the $\beta_n \asymp \sqrt{\log n}$ scaling for softmax attention when the raw attention scores $a_{1},\dots,a_{n}$ are modeled as independent $\mathcal{N}(0,1)$ random variables.

The purpose is to contrast the behavior of this Gaussian setting with the geometric setting analyzed in the main text, where pairwise score gaps remain $O(1)$ and the critical scale becomes $\beta_n \asymp \log n$.

### **Clarified the adaption of theoretical analysis in this paper to more practical transformer models**

Based on reviewers' comments on Assumption 1, we emphasized that Assumption 2 and 3 allow perturbations of the ideal simplex geometry in Assumption 1. The *scaling* remains the same in all three assumptions: the critical order $\beta_n \asymp \log n$ does not change. Assumptions 2 and 3 are therefore meant to show that the idealized Assumption 1 is already sufficient to predict the correct scaling behavior in more realistic configurations. Moreover, we added Remark 2.2 to discuss the generalization of our theoretical analysis to allow multiple exact ties and near ties in the top scores, where the same critical scaling $\beta_n \asymp \log n$ continues to hold. In our replies to the reviewers, we also clarified that our theoretical analysis works for multi-head transformer models, and for positions of tokens concentrating on several clusters.

###  **Expanded the comparison with the over-smoothing in graph neural networks (GNNs) and the attention sink in transformer models**

Based on reviewers' comments on potential connections between the phase transition phenomena in this paper and the over-smoothing in GNNs and the attention sink in transformer models, we added Section 4 to compare this paper with over-smoothing and attention sinks. Specifically, the over-smoothing in GNNs is *depth-driven*: the network dynamics can be viewed as a multiplicative composition of fixed matrices, largely insensitive to the geometric arrangement of the input features. In contrast, self-attention dynamics exhibit *length-driven* collapse, where the rank collapse can occur within a single layer and depends strongly on token positions and the context length. Second, attention sinks introduce additive stabilizing tokens to prevent excessive mixing, whereas our results focus on a multiplicative mechanism with temperature rescaling. Both approaches aim to maintain meaningful attention patterns at long but finite context lengths, and the works in attention sinks address rank collapse from distinct and potentially complementary angles.

# Final Remarks

Reviewers' comments greatly helped authors improve the clarity and quality of the paper. We add Remark 2.2 under our Assumption 2 to clarify the robustness of our assumptions and the possible adaption of the theoretical analysis in this paper to more practical transformer models. The revised manuscript further incorporates a new appendix delineating the distinctions between the $\log n$ scaling in our work and the $\sqrt{\log n}$ scaling in prior literature, alongside a dedicated section exploring links to over-smoothing in GNNs and attention sinks in transformers.

We hope this summary helps the new Area Chair in evaluating the updated submission, and we extend our sincere thanks to all reviewers for their insightful feedback.

---

### Meta-Review · Area_Chair_a9mX · 2026-01-06

**Summary:**

Overall, the reviews were fairly positive (4,6,6,8) with reviewers appreciating the theoretical analysis provided by the paper along with the accessibility of the results along with some potential practical insight in how to appropriately scale self-attention operators as context length grows.

Primary criticisms stem for questions regarding comparisons to prior work, limitations of the assumptions in the theoretical analysis, and how the theoretical results are evaluated experimentally.

**Reviewer Concerns:**

Overall, the authors appear to have responded well to the reviewers' concerns.  While reviewer concerns regarding the strength of some of the assumptions (namely that the query and key matrices are identity matrices and the lack of learned matrices) are certainly valid critiques, as is often the case with theoretical analysis simplifying assumptions are often needed for a tractable result, and many of the reviewers emphasize an interesting contribution despite these limitations.  Moreover, in their response, the authors provide some guidance about how similar results can be obtained under more general settings (such as for general Q, K matrices) but chose not to to simplify the core message.

**Reviewer Scores:**

The only reviewer who was able to reply post-rebuttal notes maintaining their positive score.  In my view, the authors have responded well to the bulk of the issues raised by the reviewers, and given the overall positive sentiment in a majority of the reviews, I would expect that ultimately a consensus to accept the paper would be reached.

---

### Decision · Program_Chairs · 2026-01-26

Accept (Poster)